# Directed-evolution of translation system for efficient unnatural amino acids incorporation and generalizable synthetic auxotroph construction

Hongxia Zhao[1,2,3,5], Wenlong Ding[1,2,3,5], Jia Zang[1,2,3], Yang Yang[4], Chao Liu[1,2,3], Linzhen Hu[1,2,3], Yulin Chen [1,2,3], Guanglong Liu[1,2,3], Yu Fang[1,2,3], Ying Yuan [2] & Shixian Lin [1,2,3✉]

Site-specific incorporation of unnatural amino acids (UAAs) with similar incorporation efficiency to that of natural amino acids (NAAs) and low background activity is extremely valuable for efficient synthesis of proteins with diverse new chemical functions and design of various synthetic auxotrophs. However, such efficient translation systems remain largely unknown in the literature. Here, we describe engineered chimeric phenylalanine systems that dramatically increase the yield of proteins bearing UAAs, through systematic engineering of the aminoacyl-tRNA synthetase and its respective cognate tRNA. These engineered synthetase/tRNA pairs allow single-site and multi-site incorporation of UAAs with efficiencies similar to those of NAAs and high fidelity. In addition, using the evolved chimeric phenylalanine system, we construct a series of *E. coli* strains whose growth is strictly dependent on exogenously supplied of UAAs. We further show that synthetic auxotrophic cells can grow robustly in living mice when UAAs are supplemented.

[1] Zhejiang Provincial Key Laboratory for Cancer Molecular Cell Biology, Life Sciences Institute, Zhejiang University, Hangzhou 310058, China. [2] Department of Medical Oncology, The Second Affiliated Hospital, Zhejiang University School of Medicine, Hangzhou 310058, China. [3] Cancer Center, Zhejiang University, Hangzhou 310058, China. [4] School of Chemistry and Chemical Engineering, Nanjing University, Nanjing 210046, China. [5]These authors contributed equally: Hongxia Zhao, Wenlong Ding. ✉email: sxlin@zju.edu.cn

The expansion of the genetic code by incorporating unnatural amino acids (UAAs) with diverse functional groups has enabled the synthesis of proteins with enhanced or novel functions[1–7]. For example, site-specific incorporation of biophysical probes, bioorthogonal handles, cross-linkers, cage groups, and natural post-translational modifications into proteins has been successfully used for a wide range of fundamental studies and advanced applications[8–16]. In this context, it's critical to have an in vivo site-specific UAA incorporation system with comparable efficiency and similar protein yield (or in this study referred to as wild-type like efficiency) when UAAs are used in place of natural amino acids (NAAs). Having a translation system where incorporating UAAs with wild-type like efficiency and low background activity in the absence of UAAs is not only very useful for the production of new proteins and biomaterials, but also for the construction of UAA-dependent synthetic auxotrophs (Fig. 1)[6,17–20]. Synthetic auxotroph that grows in the dependency of UAAs is a very promising strategy for the containment of genetically modified organisms (GMOs) and the development of live-attenuated vaccines[17,19–22].

However, introducing UAAs with wild-type like efficiency is very challenging. These engineered orthogonal aminoacyl-tRNA synthetase (aaRS)/tRNA pairs for UAA incorporation generally shows 2–3 orders of magnitude lower amino acid acylation activity compared to wild-type aaRS/tRNA pairs[18,23]. Therefore, engineered aaRS/tRNA pairs with such low activity has difficulty competing with the endogenous protein termination machinery at the amber codon (the codon commonly used for UAA incorporation). Thus, this process inevitably leads to inefficient incorporation of UAAs and truncated proteins. Despite exciting progress towards improving amino acid acylation activity, the identification of orthogonal translation systems that incorporate UAA with wild-type like efficiency and low background activity remains an outstanding challenge[24,25]. The chimeric translation systems, by hybridizing the pyrrolysyl-tRNA synthetase (PylRS)/tRNA pairs and canonical synthetase/tRNA pairs, are broadly orthogonal in prokaryotes and eukaryotes. Combining the orthogonality of the PylRS system with the diverse active sites of other canonical aaRSs, the chimeric aaRSs provide an advantage for the incorporation of UAAs that diverge from those that PylRS can incorporate in the E. coli—mammalian shuttle system[26]. In the case of the chimeric phenylalanine (Phe) system, for example, a group of Phe, tyrosine (Tyr), and tryptophan (Trp) analogs have

been successfully installed in both E. coli and mammalian cells[26]. Although genetic code expansion through broadly orthogonal systems (e.g., the pyrrolysine system and the chimeric system) has profoundly advanced our ability to incorporate diverse UAAs into a wide range of model organisms[27–31]. The incorporation of diverse UAAs with wild-type like efficiency using these broadly orthogonal systems remains largely elusive.

Here, we engineer the chimeric Phe systems with improved incorporation efficiency that achieved wild-type like efficiency in the production of full-length proteins and have low background activity in the absence of UAAs (Fig. 1). These desirable characteristics were demonstrated with multiple Phe and Trp analogs by systematically engineering the chimeric Phe tRNA (chPheT) at the acceptor stem and directed-evolution of the chimeric Phe synthetase (chPheRS) (Fig. 2a). The evolved chimeric Phe systems show single-site and multi-site UAAs incorporation into several functional proteins with wild-type like efficiency and extremely high fidelity. Impressively, we find that the engineered chPheRS/chPheT pairs showed a 65.3-fold and 22.9-fold increase in amber suppression efficiency in the presence of 4-Azido-Phe (AzF) or 3-Benzothienyl-Ala (BTA), respectively. The engineered chPheRS/chPheT pair for AzF incorporation was then introduced into a commercial E. coli strain to construct an UAA-dependent synthetic auxotroph. By introducing in-frame amber codons into the essential genes, such as dnaN, adk, tyrS, and pgsA, we were able to identify several synthetic auxotroph strains that grew robustly in the presence of AzF. In the absence of AzF, the auxotrophic strain harboring four amber codons in the dnaN gene or three amber codons in the adk gene exhibited an undetectable escape frequency on agar plates for up to 14 days (Fig. 1). Furthermore, we demonstrate that the auxotrophic strain can grow efficiently in living mice in the dependence on exogenous supply of AzF for in vivo application. Therefore, our work provides a general strategy for in vitro and in vivo application of UAA-dependent synthetic auxotrophs.

## Results

**Evolution of chimeric tRNAs.** To improve the activity of the chimeric system, we began by engineering the chimeric tRNA. In prior work, we have shown that mutation of a single base-pair in the tRNA acceptor stem can improve the amber suppression efficiency of the chimeric system[26]. Here, we sought to evolve the

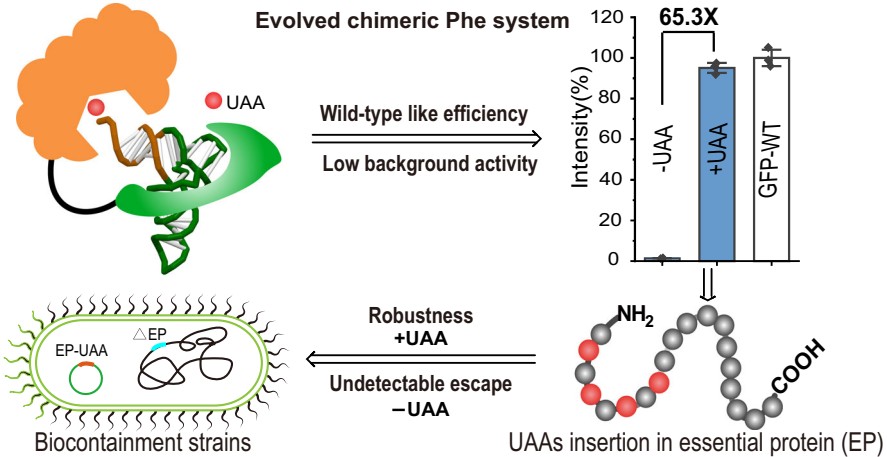

**Fig. 1 Efficient unnatural amino acids incorporation and generalizable synthetic auxotroph construction with the evolved chimeric Phe system.** Schematic illustration of the engineering of orthogonal translation systems with wild-type like efficiency in the presence of UAA and low background activity in the absence of UAA. The orthogonal translation system is used to construct UAA-dependent synthetic auxotroph by inserting amber codons into the essential gene. EP refers to essential protein.

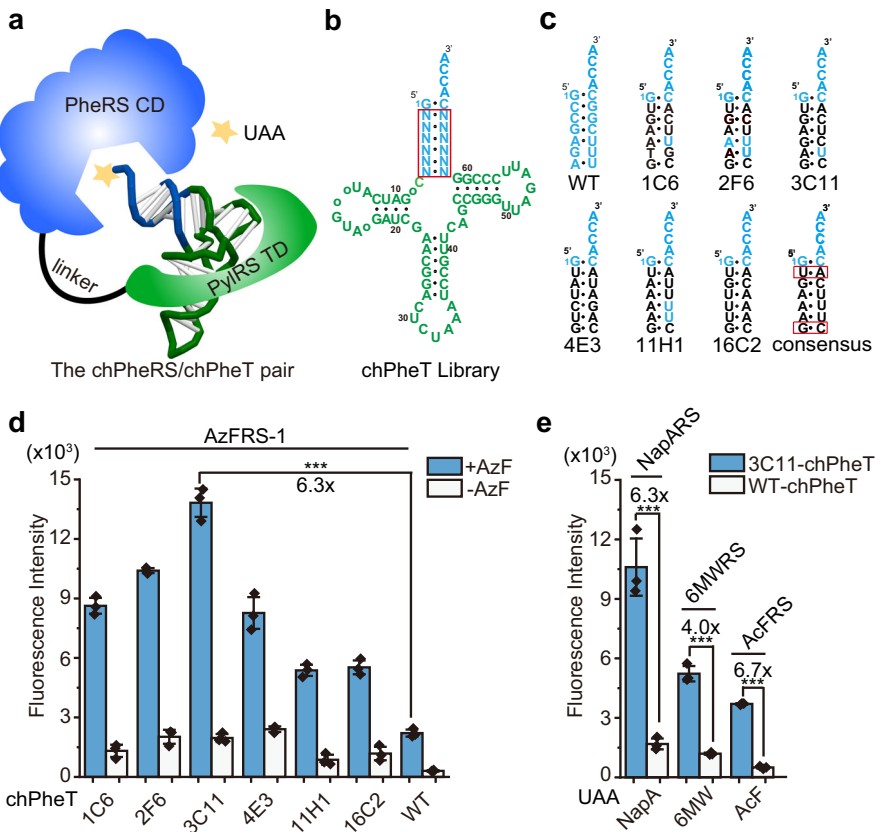

**Fig. 2 Evolution of chimeric tRNAs. a** The cartoon structure of the chimeric Phe system with components from the pyrrolysine system colored in green and components from the Phe system colored in blue. **b** The cloverleaf structure of chPheT. The nucleotides in the acceptor stem were randomized and highlighted by the red box. N represents A, T, G, or C. **c** The acceptor stem of chPheT variants with nucleotides from the WT colored in blue. The consensus chPheT is shown with the common pair-bases in all variants highlighted by red boxes. **d** Amber suppression efficiency of chPheT variants by the AzFRS-1 is tested by a GFP reporter assay with (blue bars) or without (white bars) the addition of AzF. WT-chPheT is used as the control. **e** Amber suppression efficiency of various chPheRSs with 3C11-chPheT (blue bars) or WT-chPheT (white bars) in the presence of 1 mM of indicated UAAs. Error bars represents ± standard error of the mean from three biologically independent experiments. Statistical significance is quantified with one-way ANOVA (***$p < 0.001$). Source data are provided as a Source Data file.

entire acceptor stem of the tRNA to isolate the most active tRNA variants[32]. Using wild-type (WT)-chPheT as an example, starting with the second to seventh base-pair of the chPheT acceptor stem, bases were randomized to all possible combinations to generate a library size of ~$1.7 \times 10^7$ variants (Fig. 2b). This subsequent tRNA library was subjected to one round of positive selection to identify orthogonal tRNA that showed improved amber suppression efficiency compared to the progenitor chPheT (Supplementary Fig. 2a). In the positive selection, the tRNA library was introduced to pNEG vector that carries two reporter genes, CAT-112TAG and GFP-190TAG. The library was transformed into *E. coli* DH10B competent cells harboring the AzFRS-1 (Supplementary Table 1) in the presence of AzF. The survival clones with relatively stronger fluorescence on each plate were selected. The full-length GFP fluorescence of these clones was further examined by fluorometer in the presence and absence of AzF. After screening 1,800 clones and subsequently sequencing 200 clones with significantly stronger fluorescent signals than the WT-chPheT group, several orthogonal tRNAs showed at least two-fold increase in amber suppression efficiency compared to the progenitor chPheT (Fig. 2c, d and Supplementary Fig. 2a). The acceptor stems of these newly evolved orthogonal tRNAs share some common features, all using U-A as the second base-pair and G-C as the seventh base-pair (Fig. 2c). The most active variant, 3C11-chPheT, exhibited 6.3-fold enhanced activity in the presence of AzF and low background activity in the absence of AzF

(Fig. 2d). Additionally, 3C11-chPheT was paired with various chPheRSs to access their recognition to 6-Methyl-Trp (6 MW), 2-Naphthyl-Ala (NapA), and 4-Acetyl-Phe (AcF) (Supplementary Fig. 1 and Supplementary Table 1), the amber suppression efficiency of various chPheRSs paired with 3C11-chPheT or WT-chPheT was determined by monitoring the expression of full-length GFP-190TAG in the presence of the corresponding UAAs. The results showed a 4.0–6.7-fold increased activity in incorporating these UAAs when 3C11-chPheT was used compared to WT-chPheT (Fig. 2e), indicating that 3C11-chPheT generally enhanced the activity of the chimeric Phe system. Furthermore, the amber suppression assay in mammalian cells demonstrated that 3C11-chPheT was active and orthogonal in eukaryotic cells as well (Supplementary Fig. 3b).

**Evolution of chimeric aaRSs.** We next focused on the directed-evolution of chPheRS to enhance its enzymatic activity. The chPheRS consists of an N-terminal tRNA binding domain that deviates from PylRS and a C-terminal amino acid recognition domain that deviates from human mitochondrial PheRS[26]. Given that the N-terminal domain of chPheRS has been engineered previously[33,34] (Fig. 2a), we reasoned that mutants of the C-terminal domain of chPheRS were a more promising option for the subsequent directed-evolution process compared to the full-length protein. Therefore, we diversified the C-terminal domain of AzFRS-1 (Supplementary Table 1) by multiple rounds of error-

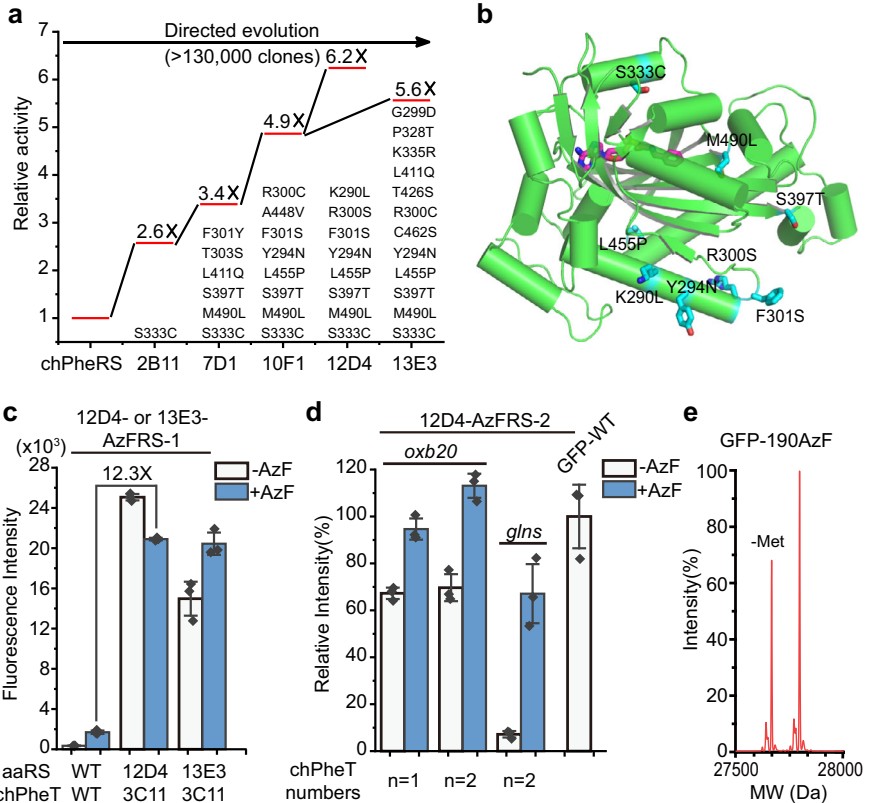

**Fig. 3 Evolution of chimeric aaRSs. a** Directed evolution of the chPheRS for efficient genetic code expansion. The evolution trajectory of representative clones shows increased amber suppression activity compared to the progenitor chPheRS. Mutations in these representative clones are shown. **b** Structure of C-terminal domain of human mitochondrial PheRS, in which mutation residues in clones 12D4 are shown in sticks and phenylalanyl-adenylate in the amino acid binding pocket is highlighted in magenta. **c** Amber suppression efficiency of the evolved 12D4- or 13E3-AzFRS-1/chPheT pairs is tested by the GFP reporter assay. **d** Amber suppression efficiency of the evolved 12D4-AzFRS-2/chPheT pairs is tested, in which the 12D4-AzFRS-2 is under the control of *oxb20* or *glns* promoter, respectively. **e** Mass spectrometry characterization of the fidelity of AzF incorporation into GFP with the 12D4-AzFRS-2/3C11-chPheT pair. The expected molecular weight (MW) 27796 Da; observed 27797 Da. Error bars represents ±standard error of the mean from three biologically independent experiments. Source data are provided as a Source Data file.

prone polymerase chain reaction (PCR) with a mutation rate of $1.6 \times 10^{-2}$, followed by positive selections (Supplementary Fig. 2b). In each round of selection, the aaRS library in the pBK vector was transformed into DH10B competent cells carrying chPheT and a dual reporter system (CAT-112TAG and GFP-190TAG) in the presence of AzF. The survival clones with stronger fluorescence signal than AzFRS-1 were selected for validation and sequencing. The resulting chPheRS genes carrying prominent mutations in each round of positive selection were used as templates for the next round of diversification. Four rounds of iterative diversification were performed by error-prone PCR, followed by screening and validation of full-length GFP expression with progressively higher stringency in each round. After screening over 130,000 clones from the four rounds of directed-evolution, two dominant clones, 12D4-chPheRS and 13E3-chPheRS, were obtained (Fig. 3a). The 12D4-AzFRS-1 showed 6.2-fold higher efficiency than the progenitor chPheRS, while 13E3-AzFRS-1 displayed 5.6-fold higher efficiency. Interestingly, they share several mutations in common (Fig. 3a). Importantly, none of these mutations appear in the amino acid binding pockets, suggesting that these mutations can be transferred to other chPheRSs to efficiently recognize various UAAs (Fig. 3b).

**Efficient single-site incorporation of various UAAs.** We envisioned that the combination of the evolved chPheT and chPheRS should further increase the activity of the chimeric Phe system. As

expected, GFP reporter assay showed a dramatic increase in the amber suppression efficiency when combining the most active 12D4-chPheRS/3C11-chPheT pair (Fig. 3c). The efficiency of the most active pair was ~12.3-fold higher than the progenitor chPheRS/chPheT pair. A similar result was detected when using the 13E3-chPheRS/3C11-chPheT pair (Fig. 3c). Since high background activity of 12D4-AzFRS-1 towards Trp was detected in the absence of AzF (Fig. 3c and Supplementary Fig. 5b), the amino acid binding pocket of chPheRS was then altered to improve its specificity to AzF and lower its specificity to Trp. Briefly, we generated a library of chPheRS variants from WT-chPheRS by saturated mutagenesis of residues (F464, T467, and A507) that may interact with AzF based on the crystal structure in order to identify the most active AzFRS in the library. The generated chPheRS library was subjected to subsequent positive and negative selections. Positive selection was performed to enrich functional chPheRSs that read through dual reporter genes in the presence of AzF. Negative selection was conducted to exclude chPheRSs recognizing 20 NAAs. After two rounds of positive selection, the survived clones exhibiting a stronger GFP fluorescence signal were selected and subsequently sequenced to isolate prominent chPheRS variants that recognized AzF. We were able to obtain 12D4-AzFRS-2 (carrying the mutations: F464I, T467G, and A507G) that incorporated AzF with wild-type like efficiency and much lower background activity compared to 12D4-AzFRS-1 (Fig. 3d). Moreover, the amber suppression efficiency was further enhanced with the expression of two copies of

3C11-chPheTs (Fig. 3d). Surprisingly, we observed that the expression level of GFP-190AzF was surpassing that of WT-GFP-190Asp (Fig. 3d). Moreover, the evolved chimeric Phe system produced undetectable levels of truncated proteins in the presence of UAAs, testifying that the most active chimeric Phe pair outcompeted the endogenous protein termination machinery at the amber codon (Supplementary Fig. 4). This is consistent with previous discoveries that the expression level of UAA-containing proteins can surpass that of their WT proteins when a highly efficient translation system is introduced[18,35]. Besides, we detected extremely high fidelity of AzF incorporation (>99%) by LC-MS analysis (Fig. 3e and Supplementary Fig. 5). Similarly, in additional to site 190 on GFP, the randomly selected sites on GFP (sites: 140, 151, 172, 173, 182, and 193) were detected with wild-type like or surpassing efficiency and excellent incorporation fidelity using the 12D4-AzFRS-2/3C11-chPheT pair (Supplementary Fig. 5 and Supplementary Table 1). Notably, excellent incorporation efficiency in these selected sites on GFP was also detected using the AzFRS-2/3C11-chPheT pair with extremely low background activity (Supplementary Fig. 6 and Supplementary Table 1). Therefore, we hypothesized that high background activity was likely generated by plasmid-based overexpression of the highly active 12D4-AzFRS variants. Indeed, when 12D4-AzFRS-2 was integrated into the chromosome, background activity was significantly reduced and excellent amber suppression efficiency was observed (Supplementary Fig. 7). Similarly, when the expression of 12D4-AzFRS-2 was driven by the mild glutaminyl-tRNA synthetase (glns) promoter rather than the strong promoter oxb20 on the plasmid, background activity of 12D4-AzFRS-2 was significantly reduced and high amber suppression efficiency (~70% of WT protein) was still observed (Fig. 3d). Thus, using chromosomally integrated aaRS, a weaker aaRS promoter, and/or wild-type AzFRS instead of 12D4 version, background activity can be significantly reduced (Fig. 3d and Supplementary Figs. 6–7). Furthermore, time-course analysis of GFP-190TAG and GFP-2*TAGs using the 12D4-AzFRS-2 revealed a significantly reduced rate of protein production in the absence of AzF in living cells (Supplementary Fig. 8). Together, these results demonstrate that 12D4-chPheRS kinetically favors AzF over NAAs.

To examine whether the evolved chimeric Phe system affects the robustness of host cells, DH10B cells were transformed with a vector carrying 3C11-chPheT and 12D4-AzFRS-2 to assess the effect of background incorporation in the host. The growth rate assay revealed that the fitness of the transformed DH10B was not impaired when the evolved chimeric Phe system was introduced compared to WT-DH10B cells (Supplementary Fig. 9).

Because the beneficial mutations in 12D4-chPheRS and 3C11-chPheT do not appear in the amino acid binding pocket, we hypothesized that the improved amino acid acylation activity of this evolved system should be translated to other UAAs by their cognate chPheRSs. Impressively, five additional UAAs, including AcF, NapA, 6 MW, BTA, and 7-Methyl-Trp (7 MW) (Supplementary Fig. 1), can be incorporated with wild-type like efficiency by their cognate 12D4-chPheRSs (Supplementary Table 1), attesting to the high incorporation efficiency of the evolved chimeric system (Fig. 4a). High fidelity of incorporation (>99%) was confirmed by LC-MS analysis (Fig. 4b), further demonstrating that 12D4-chPheRSs favored UAAs over NAAs. These UAAs carrying bioorthogonal handles and biophysical moiety have been successfully used in protein bioconjugation and tailoring protein function[32,36,37]. Among them, NapARS, 6MWRS, and 7MWRS were reported in our previous study[26]. The AzFRS-1 showed a poly-specificity to AcF, and 7MWRS showed a poly-specificity to BTA (Supplementary Table 1). Together, these results demonstrate that the evolved chimeric Phe systems enable the

incorporation of various UAAs with wild-type like efficiency in commercial E. coli strains and with high incorporation fidelity and robust cellular fitness.

In addition, to benchmark the efficiency of our newly evolved chimeric system, we decided to incorporate UAA into other functional proteins that are more difficult to amber-suppress than the commonly used GFP. We selected eight additional proteins in the laboratory, including ubiquitin conjugating enzyme E2 K (UBE2K), adenylate kinase (ADK), tyrosyl-tRNA synthetase (TyrRS), firefly luciferase (Fluc), GID complex subunit 4 (GID4), PHD domain of BPTF protein (PHD), synthetic cytokine neoleukin-2/15 (Neo-2/15)[38], and ubiquitin (Ub), and randomly chosen amber codons on these proteins for AzF incorporation. Strikingly, as indicated by the purified protein yields, the incorporation of AzF into these proteins was detected with similar incorporation efficiency to NAAs into the wild-type proteins (Fig. 4c). And high fidelity of AzF incorporation in these proteins was verified by LC-MS analysis (Supplementary Fig. 10). These proteins span many protein classes, including therapeutic protein, enzyme, metalloprotein, and signaling protein. Therefore, the newly evolved chimeric system should have very broad applications in the synthesis of various UAA-containing proteins in vitro and in vivo.

Moreover, since the chimeric Phe system is an E. coli–mammalian shuttle system, the evolved system can be directly transferred to mammalian cells. Indeed, we found that the two most active chPheRS/chPheT pairs showed significantly increased activity compared to the progenitor pair in the mammalian system (Supplementary Fig. 3c). In addition, the activity of the most active chPheRS/chPheT pair was higher than the gold-standard Boc-Lysine synthetase/tRNA pair in mammalian cells[34] (Supplementary Fig. 3).

**Efficient multi-site UAA incorporation.** With the efficient chimeric translation system in hand, we then tested whether the most active 12D4-chPheRS/3C11-chPheT pair could improve the incorporation efficiency of multi-site UAAs to wild-type like level. We used a genomically recoded organism (GRO) to express a protein carrying multiple UAAs. In this GRO (C321.ΔA exp strain), all 321 amber stop codons were replaced with synonymous ocher codons and the release factor 1, which recognizes amber codons, was deleted[17]. This GRO has been successfully expressed proteins that have multiple UAAs incorporated[18–20]. We constructed a GFP control plasmid in which the elastin-like polypeptide (ELP) was fused to the N-terminus of the GFP gene[18]. Meanwhile, experimental construct, ELP-GFP, that carry three amber codons was constructed as readout (Fig. 4d). Each of these constructs were co-transformed into GRO with the 12D4-AzFRS-2/3C11-chPheT pair to assess the expression of full-length ELP-GFP. The amber suppression assay in GRO showed that the evolved chimeric Phe pair can incorporate three-site AzFs into ELP-GFP with wild-type like efficiency (Fig. 4d). Additionally, the proteins were purified using the C-terminal affinity tag. The yield of purified protein carrying three-site AzFs was 100 mg/L, in contrast 75 mg/L for the purified WT protein (Fig. 4d). Similarly, we detected high incorporation fidelity of multi-site AzFs by LC-MS (Supplementary Fig. 11). These data demonstrate that the evolved chimeric Phe system allows multi-site UAAs incorporation with wild-type like efficiency in the GRO. Taken together, we concluded that the broadly orthogonal chimeric Phe system is capable for single-site and multi-site UAA incorporation with wild-type like efficiency and extremely high fidelity.

**Construction of UAA-dependent synthetic auxotroph.** We then investigated whether the evolved chPheRS/chPheT pairs can be

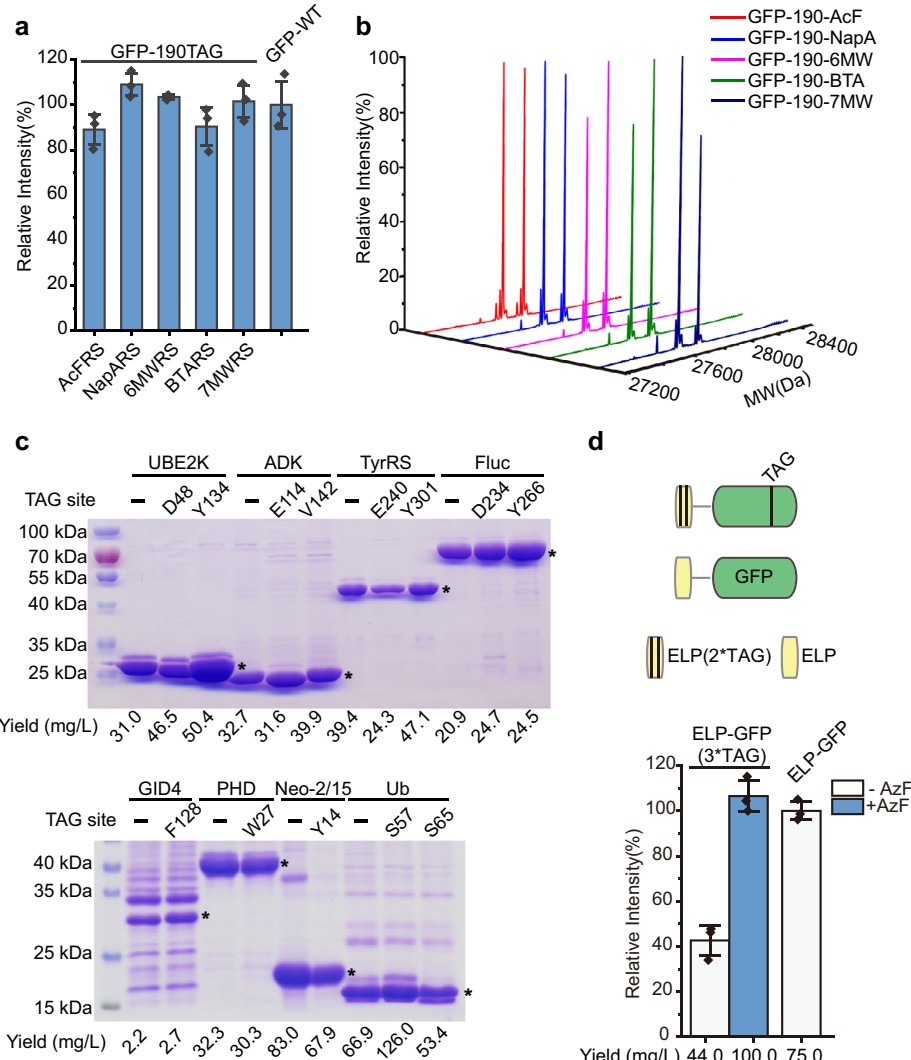

**Fig. 4 Single-site and multi-site incorporation of UAAs. a** Amber suppression efficiency of the indicated 12D4-chPheRS variants is tested in the presence of their cognate UAAs. The detailed mutation information is shown in Supplementary Table 1. **b** Mass spectrometry characterization of the fidelity of UAA incorporation into GFP with the indicated 12D4-chPheRS variants. The expected MW from left to right are 27797, 27805, 27808, 27812, and 27808 Da, and the observed MW are 27798, 27806, 27809, 27811, and 27809 Da. In each spectrum, N'-Met cleaved peaks are also detected. **c** Analysis of amber suppression activity by the evolved 12D4-AzFRS-2/3C11-chPheT pair at the indicated position of eight proteins by detecting expression of full-length protein using wild-type proteins as positive controls. The major bands indicated by stars in Coomassie blue staining gel are the corresponding full-length proteins. The purified protein yield is shown below the gel for side-by-side comparison. The experiment in the figure was repeated twice with similar results. **d** Schematic illustration of reporter ELP-GFP, in which a black line indicates one amber codon. WT-ELP-GFP is used as the control. Amber suppression efficiency of the evolved chimeric Phe system incorporates three-site AzFs into ELP-GFP. The purified protein yield is shown below the figure. Error bars represents ±standard error of the mean from three biologically independent experiments. Source data are provided as a Source Data file.

used to construct UAA-dependent synthetic auxotrophic strains that are strictly dependent on exogenously supplied UAAs for growth. The construction of UAA-dependent synthetic auxotrophs has provided a robust strategy for the containment of GMOs and the development of safe live-attenuated vaccines[19,20]. Decoding amber codons with UAAs by the orthogonal aaRS/tRNA pair is a crucial prerequisite for the construction of such synthetic auxotrophs. Synthetic auxotrophs dependent on the addition of UAAs can be constructed by introducing single- or multi-site amber codons into the essential genes, as cell growth requires the expression of these essential proteins. In this context, several strategies have been reported to construct UAA-dependent synthetic auxotrophs with low escape frequencies[19,20,39–43]. Despite exciting progress, simple and general strategies to efficiently construct UAA-dependent common bacterial strains with very low escape frequency remain to be an outstanding challenge[42,44,45].

And the utility of UAA-dependent synthetic auxotrophs in living animals is almost unknown. Additionally, previous studies have relied primarily on the *Methanococcus Jannaschii* (*Mj*) TyrRS/tRNA pair, which is not orthogonal in eukaryotes, limiting their potential applications in the design of synthetic eukaryotic auxotrophs. Therefore, we envisioned that the engineered chimeric Phe system with high efficiency and broadly orthogonal in prokaryotes and eukaryotes should largely expand our ability to design UAA-dependent synthetic auxotrophs.

To generate a UAA-dependent essential protein, the aaRS/tRNA pair that can exclusively incorporate UAAs at the amber codon is needed. Ideally, the aaRS/tRNA pair would produce a sufficient amount of UAA-containing essential protein to allow robust growth of synthetic auxotroph with the addition of UAAs, in contrast to no expression of full-length essential protein in the absence of UAAs. Unfortunately, all orthogonal translation

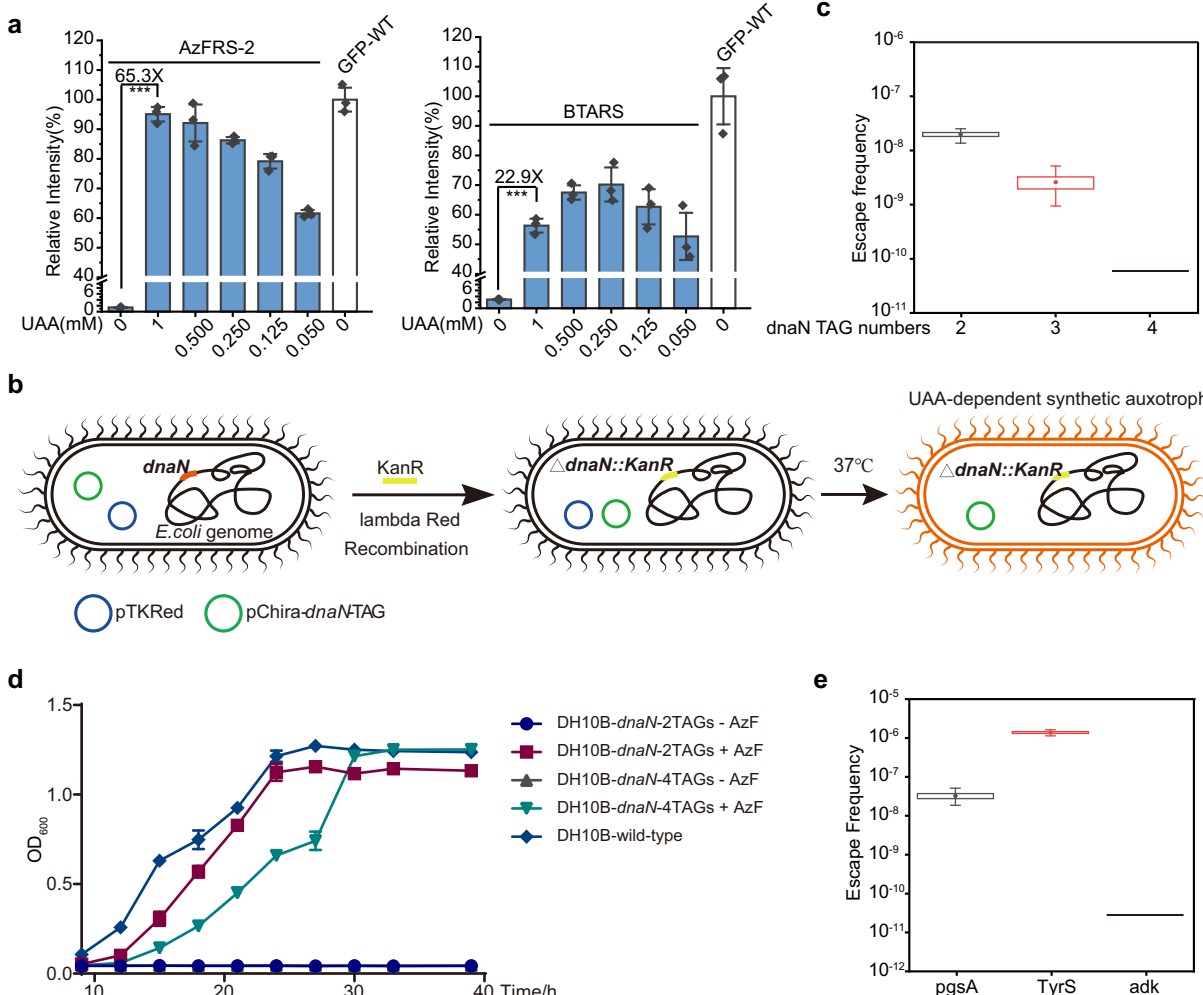

**Fig. 5 Construction of UAA-dependent synthetic auxotroph. a** Amber suppression efficiency of the AzF system and the BTA system is tested in the presence of indicated concentration of AzF or BTA. Statistical significance is quantified with one-way ANOVA (***$p < 0.001$, $p = 5.74E–7$ and $4.24E–6$, respectively). **b** Schematic illustration of the procedure of engineering UAA-dependent synthetic auxotroph. The pTKRed vector carries the lambda Red recombinase system, and the pChira vector carries AzFRS-2 (under the control of *oxb20* promoter), 3C11-chPheT (under the control of *lpp* promoter), and the *dnaN*-TAG gene (under the control of *tac* promoter). **c** Escape frequencies of *E. coli* strains at day 14 post-plating with indicated numbers of TAG in the *dnaN* gene. The black line represents escape frequency below the detection limit. **d** Growth curve of various UAA-dependent synthetic auxotrophs in the presence or absence of AzF. **e** Escape frequencies of *E. coli* strains at day 14 post-plating with three TAGs in the indicated genes. The black line represents escape frequency below the detection limit. Error bars represents ±standard error of the mean from three biologically independent experiments. The box plots showing the minima, maxima, center, bounds of box and whiskers and percentiles. Source data are provided as a Source Data file.

systems showed background amber suppression activity[19,20,39–43] due to the activity of aaRS to 20 NAAs and the incomplete orthogonality of tRNA. The amber suppression efficiency of various chimeric Phe systems in our study was examined by monitoring the expression of full-length GFP-190UAA, in order to gauge the dynamic range of gene expression in the presence or absence of their cognate UAAs. After screening all the available chPheRS/3C11-chPheT pairs in our study, we were able to obtain two systems that showed a very wide dynamic range of gene expression in the presence or absence of UAAs. One was the AzF system introduced by the AzFRS-2/3C11-chPheT pair and the other was the BTA system introduced by the BTARS/3C11-chPheT pair (Fig. 5a and Supplementary Fig. 12). The results showed that both the AzF system and the BTA system increased the expression of GFP-190UAA by 65.3-fold or 22.9-fold, respectively. Since the AzF incorporation has been systematically improved by the *Mj*.TyrRS system[18,46], we compared the efficiency and background activity of our AzF system side-by-side with the previously engineered AzF system (Supplementary

Fig. 13). Our system was the only one that exhibited wild-type like efficiency in the presence of AzF and extremely low background activity in the absence of AzF. Besides, GFP fluorescence analysis revealed that the AzF system was able to produce GFP efficiently, with only 9.3% or 35.2% loss of protein yield with a 4-fold or 20-fold reduction in AzF concentration, respectively. Impressively, the BTA system produced more or similar amounts of full-length GFP with a 4-fold or 20-fold reduction in BTA concentration (Fig. 5a and Supplementary Fig. 12). To our knowledge, such a wide dynamic range of gene expression has never been observed in the literature using the *Mj*.TyrRS system. Furthermore, our evolved AzF system enabled incorporate three-site AzFs into ADK and DnaN proteins with much higher efficiency than the *Mj*.AzFRS system and undetectable background activity in DH10B cells (Supplementary Fig. 14), allowing fine-tuning the expression of essential protein in the AzF-dependent manner. Collectively, these results demonstrate that the chimeric Phe system is a very promising system in the construction of synthetic auxotrophs. Since the AzF system

displays a greater dynamic range of gene expression and it has been successfully used in expanding the genetic code of large model animals, we set out to extend the application of synthetic auxotroph to living animals using the AzF system.

To construct an AzF-dependent synthetic auxotroph, we planned to insert an in-frame amber codon (from two to four amber codons) into the essential gene, *dnaN*, encoding a subunit of the DNA polymerase III holoenzyme[42,47]. The chromosomal *dnaN* gene was knocked out using the lambda Red recombinase system encoded in the pTKRed vector (Fig. 5b). DH10B cells were co-transformed with pTKRed and pChira vector carrying AzFRS-2, 3C11-chPheT, and *dnaN*-TAG gene. The AzFRS-2 gene is under the control of the constitutive *oxb20* promoter, whereas the *dnaN*-TAG gene is driven by the constitutive *tac* promoter. Therefore, the expression of full-length DnaN protein is controlled by the addition of AzF alone, which simplifies its usage in vitro and in vivo. A single clone carrying both plasmids was picked and competent cells were prepared for subsequent transformation of the KanR gene with homology arms to replace the chromosomal *dnaN* gene (Fig. 5b). The knockout clones were verified by PCR, followed by removal of the pTKRed vector carrying a temperature-sensitive replication origin (Fig. 5b). After screening, we successfully replaced the chromosomal *dnaN* with *dnaN* expressed by plasmids that carries two, three, or four in-frame amber codons, respectively (Supplementary Fig. 15a). To test whether these cells were grown in an AzF-dependent manner, these cells were cultured on LB-agar medium with or without AzF. Most of the cells grew much faster in the presence of AzF, suggesting that the AzF-dependent synthetic auxotroph had been successfully generated (Supplementary Fig. 15b). Three clones from each group were selected to measure escape frequencies. These auxotrophic cells were plated on agar plates without AzF and colony forming units (c.f.u) were measured after two weeks of incubation. The escape frequencies of cells carrying *dnaN*-2*TAGs, −3*TAGs, and −4*TAGs genes were $2.0 \times 10^{-8}$, $2.6 \times 10^{-9}$, and $<6.0 \times 10^{-11}$, at day 14 after plating, respectively (Fig. 5c and Supplementary Fig. 16). As the number of TAGs in the *dnaN* gene increased, progressively lower escape frequency was detected. And the escape frequency of strains carrying three or four amber codons reached the National Institute of Health recommended requirement of $10^{-8}$ [48]. Notably, no escape clones were detected in the four amber codons group up to 14 days. Besides, growth curve assays showed that these auxotrophic cells grew robustly in the presence of AzF in the liquid media (Fig. 5d). We then sought to insert amber codons to other essential genes in DH10B cells to test the versatility of our strategy. We constructed several essential genes carrying three in-frame amber codons and assayed the escape frequency of these cells 14 days post-plating. Impressively, *adk* gene bearing three amber codons showed undetectable escape frequencies ($<1.0 \times 10^{-11}$) (Fig. 5e), demonstrating the generalizability of our strategy in construction of synthetic auxotroph in a wide range of hosts using *dnaN* or *adk* gene alone. We envisioned that the addition of more amber codons to these essential genes and/or the combination of two essential genes should further lower the escape frequency. These results demonstrate that the AzF system using the evolved chimeric Phe pair is an ideal system for the construction of UAA-dependent synthetic auxotrophs.

**UAA-dependent synthetic auxotroph in vivo.** Finally, we pursued the application of introducing a UAA-dependent synthetic *E. coli* strain with the evolved chimeric Phe system in living mice. To assay the growth of this live-attenuated *E. coli* strain in living animals, cells carrying *dnaN*-3*TAGs were transformed with a luciferase reporter gene under the control of the *trp* promoter. We

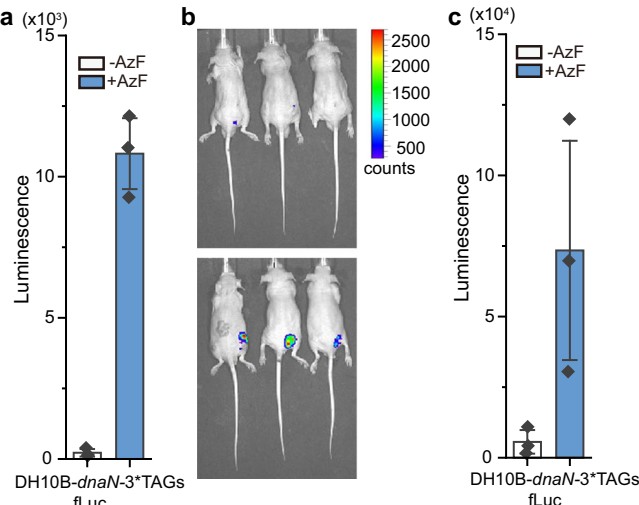

**Fig. 6 Application of UAA-dependent synthetic auxotroph in living mice. a** Bioluminescence intensity of UAA-dependent synthetic auxotroph constructed with 3TAGs in the *dnaN* gene in the presence or absence of AzF in vitro. **b** Bioluminescence imaging for UAA-dependent synthetic auxotroph in the absence (up) or presence (down) of AzF in living mice. Cells (~1 × 10⁸ cells/50 μl) were subcutaneously injected into the mice (left legs, DH10B cell; right legs, UAA-dependent synthetic auxotroph expressing fLuc protein). The bioluminescent signal was measured in whole animals at 24 h after injection. **c** Statistic analysis of bioluminescent value in **b**. Error bars represents ±standard error of the mean from three biologically independent experiments. Source data are provided as a Source Data file.

hypothesized that if cells exhibited a UAA-dependent growth in vivo, a strong luciferase signal would only be detectable in the mice with the supply of AzF. Indeed, we observed a dramatic increase in the bioluminescent signal of *E. coli* cells with the addition of AzF in vitro (Fig. 6a). The AzF-dependent auxotrophic *E. coli* cells were then inoculated into nude mouse through subcutaneous injection. The inoculated mice were supplied with AzF, while PBS buffer was supplied as a negative control. Twenty-four hours after inoculation, luciferase activity was visualized in nude mice (Fig. 6b). We detected a strong bioluminescent signal with the addition of AzF, while very low background activity was observed in the negative control group (Fig. 6b, c). These results demonstrate that the synthetic auxotrophic *E. coli* grow robustly in living mice with the supplement of UAA. Taken together, the evolved chimeric Phe system is uniquely suited to provide UAA-dependent synthetic auxotrophs that work both in vitro and in vivo for the containment of genetically modified organisms and the development of safe live-attenuated vaccines.

## Discussion

In summary, we have successfully improved the efficiency of the chimeric translation system for the incorporation of UAA by engineering the chPheT and directed-evolution of the chPheRS. The evolved chimeric Phe systems show extremely high activity, enabling single-site and multi-site incorporation of several Phe and Trp analogs with wild-type like efficiency and high fidelity. These incorporated UAAs carrying bioorthogonal handles and biophysical features have been widely used for protein bioconjugation and tailoring protein functions[32,36,37,46]. Identifying orthogonal translation systems that incorporate UAAs with wild-type like efficiency remains an outstanding challenge. Our study provides more orthogonal translation systems for efficient UAAs incorporation, which expands our ability to produce a large

number of functionalized proteins with enhanced or new functionalities. Because the chPheRS/chPheT pair is an *E. coli*–mammalian shuttle system for genetic code expansion in prokaryotes and eukaryotes, the evolved system achieves a much higher UAAs incorporation efficiency in mammalian cells compared to the progenitor system. More importantly, the directed-evolution procedures applied to the chimeric Phe system is generally applicable, we thus expect that more broadly orthogonal translation systems with extremely high activity can be engineered using similar strategies in the near future.

In addition, the engineered chimeric synthetase/tRNA system show more than 65-fold increased amber codon suppression efficiency in the presence of AzF, making it an ideal system in the construction of synthetic auxotrophs. Indeed, by inserting amber codons into the essential genes, such as *dnaN* and *adk*, we are able to identify synthetic auxotrophs that grow robustly in the presence of AzF and exhibit undetectable escape frequencies on solid media for up to 14 days. This strategy is generally applicable to other essential genes and several UAA-dependent synthetic auxotrophs are constructed. Furthermore, we show that the auxotrophic strain can grow efficiently in living mice in the dependence of the exogenously supplied UAA. Therefore, the engineered chimeric Phe systems are uniquely suited to provide safe GMOs for basic research and applied advances. More broadly, due to its broad orthogonality in both prokaryotes and eukaryotes, the chimeric Phe system paves the way for the construction a series of UAA-dependent eukaryotic cells, such as yeast and mammalian cells. Therefore, the evolved systems should greatly expand our ability to produce functionalized proteins, construct biosafety laboratory strains, and provide live-attenuated bacterial vaccines.

## Methods

**Reagents**. Unless otherwise noted, all commercial reagents were used directly without further purification. Primers and genes were synthesized by Tsingke Biotech. Anti-GFP rabbit antibody (Cell Signaling Technology, cat#2555, lot#6) was used at a dilution of 1:1000. Goat anti-rabbit IgG (H + L) HRP conjugate (Proteintech, cat#SA00001-2, lot#20000311) was used at a dilution of 1:5000.

**Instrumentations**. $OD_{600}$ and fluorescence intensity were acquired with Bio Tek Synergy NEO2. Western blotting PVDF membranes were imaged by Azure Biosystems C400. FACS data were collected by Beckman CytoFlex. LC-MS analysis was performed on an Xevo G2-XS QTOF MS System.

**Software and code**. GFP fluorescence for assessment of amber suppression efficiency were collected with Gen5 CHS 2.09 software. Growth curve of DH10B strains were collected with Gen5 CHS 2.09 software. Bioluminescence signals for monitoring the survival of UAA-dependent synthetic auxotroph in presence or absence of UAA were collected with Gen5 CHS 2.09 software in vitro, with IVIS Spectrum in living mouse. Growth curve of UAA-dependent synthetic auxotrophs were collected with spectrophotometer DU730. FACS of HEK 293T cells were acquired with CytExpert 2.0. Western blotting PVDF membranes were captured by cSeries Capture Software.

The assessment of activity for chimeric pairs were processed with Origin 8.0 software. Comparisons of activity for UAA incorporation with different orthogonal pairs were performed using one-way ANOVA with significance level at α = 0.05 in Origin 8.0 software. The growth curve of strains was processed with Origin 8.0 software. Mass spectral deconvolution was performed using UNIFI software (version 1.9.4, Waters Corporation). FACS data were processed with FlowJo (version 10, Treestar Software).

**Strains**. DH10B strain was used for protein expression and construction of synthetic auxotrophs. C321.ΔA.exp (NCBI accession number: CP006698.1) was used for protein expression with multi-site UAAs.

**Cell culture procedure**. HEK 293T cells (from ATCC) were maintained in an exponential growth as a monolayer in Dulbecco's Modified Eagle Medium (Thermo Fisher Scientific), high glucose, 10% fetal bovine serum (Thermo Fisher Scientific), 1% penicillin-streptomycin, and cultured at 37 °C in 5% $CO_2$.

**Mice**. Balb/c mice (male, 6–8 weeks) were purchased from Shanghai Model Organisms. All mice were reared in-house (temperature: 20–25 °C, humidity: 40–60%) in 12 h light/dark cycle. All animals had free access to food and sterilized water. Mice were maintained under specific-pathogen-free conditions, and all mouse experiments were approved by the Institutional Animal Care and Use Committee of Zhejiang University.

**Plasmid construction**. Unless otherwise stated, all plasmids were constructed by Gibson assembly. Plasmid pTKRed bears the pSC101 replicon, the temperature-sensitive *Rep101ts* gene, and the λ red system. Plasmid PKD4 bears a kanamycin resistance cassette. To construct a plasmid bearing dual copies of chPheT, the cassette of *lpp* promoter, chPheT and *rrnc* terminator was amplified from pNEG-GFP-190TAG-chPheT and inserted into the same plasmid to generate pNEG-GFP-190TAG-2*chPheT. To construct plasmids expressing the essential genes carrying amber codons: firstly, a pEVOL fragment containing p15A origin and chloramphenicol resistance gene, a pBK fragment containing chPheRS driven by *oxb20* promoter, and a pNEG fragment containing two copies of chPheT were amplified and assembled to yield a new plasmid dubbed as pChira-chPheRS-2*chPheT; secondly, the constitutive *tac* promoter was inserted for the expression of essential gene; finally, a synthetic sequence containing amber codons was inserted into the N-terminus of the essential genes to generate the final version of plasmid termed pChira-x*TAG-EG-chPheRS-2*chPheT (EG: The name of essential gene, x: TAG numbers, Synthetic sequence: ATGTAGGTTCCAGGTTAGGGTGGT (x = 2), ATGTAGGTTTAGGGTTAGGGTGG T (x = 3), ATGTAGGTTTAGGGTTAG GGTTAG (x = 4)). Four essential genes were selected in this study: *dnaN*, *pgsA*, *adk*, and *tyrS*. To construct a plasmid for *E. coli* chromosome integration cassette, the 300-bp chromosomal homology region at position 17231–17232 in DH10B was assembled to pChira-chPheRS-2*chPheT vector, named as pChira-chPheRS-2*chPheT-site1. The 300-bp homology region at position 2048790–2048791 in DH10B was assembled to pChira-chPheRS-2*chPheT-KanR vector, named as pChira-chPheRS-2*chPheT-site2. To construct a plasmid for bioluminescent visualization of *E. coli* in mouse, the gene of Fluc under the constitutive *trp* promoter was assembled to pGEX-6P-1 vector, giving a plasmid dubbed as pGEX-trp-fLuc.

**Assessment of amber suppression efficiency by GFP reporter assay in *E. coli***. Plasmid pNEG (carrying GFP-190TAG and chPheT) and plasmid pBK (carrying chPheRS) were co-transformed into chemically competent DH10B cells. The transformed cells were recovered in 2xYT medium with shaking for 1 h at 37 °C and plated on LB agar containing 50 μg/ml kanamycin and 100 μg/ml ampicillin for 12 h at 37 °C. A single colony was picked and grown in 2 ml of 2xYT medium containing required antibiotics at 37 °C until $OD_{600}$ reaching ~0.8. Protein expression was induced by the addition of arabinose with a final concentration of 0.2% for 16 h at 30 °C with or without the corresponding UAA. After induction, 0.75 ml of cell culture was collected by centrifugation, and then lyzed by 150 μl BugBuster Protein Extraction Reagent (Millipore) for 20 min at room temperature. The supernatant of the lysate (100 μl) was transferred to a 96-well cell culture plate (Costar). And GFP signals of the supernatant were recorded by Bio Tek Synergy NEO2 with a background subtraction and normalized by the bacterial density ($OD_{600}$) that was measured by Bio Tek Synergy NEO2 as well.

**Protein expression, purification, and LC-MS analysis**. For protein expression and purification, overnight cultured DH10B cells were diluted at a ratio of 1:100 into 100 ml of fresh LB medium supplemented with required antibiotics. Cells were grown until $OD_{600}$ reached ~0.8. L-arabinose was added with a final concentration of 0.2% to induce GFP expression (30 °C, 220 rpm, 16 h) with or without the addition of corresponding UAA. Cells were harvested by centrifuging at 3000 *g* for 5 min at 4 °C. The resulting cell pellets were suspended in ice-cold buffer A (25 mM Tris, 250 mM NaCl, pH 8.0, 2 mM β-ME) and sonicated. The suspension was centrifuged at 10,000 *g* for 60 min at 4 °C. For 6xHis tag fusion proteins, the resulting supernatant was purified via Ni²⁺-affinity chromatography on chelating Sepharose equilibrated with buffer A, and washed with 6 volumes of buffer A containing 50 mM imidazole. The proteins were eluted with buffer A supplemented with 500 mM imidazole. For GST tag fusion proteins, the resulting supernatant was purified via glutathione agarose resin. The protein was eluted with Buffer A supplemented with 20 mM reduced glutathione. Purified proteins were subjected to SDS-PAGE and LC-MS analysis. Purified protein yields were obtained by measuring the absorbance at 280 nm (395 nm for GFP variants) using a spectrometer. To evaluate the fidelity of AzF incorporation, GFP variants bearing AzF were reduced overnight by adding tris(2-carboxyethyl)phosphine (TCEP) at a final concentration of 10 mM at 4 °C. After reduction, the proteins were buffer exchanged to water using Amicon Ultra columns for the subsequent LC-MS analysis.

For LC-MS analysis, the purified proteins were analyzed on an Xevo G2-XS QTOF MS System (Waters Corporation) equipped with an electrospray ionization (ESI) source in conjunction with Waters ACQUITY UPLC I-Class plus. Separation and desalting were carried out on a Waters ACQUITY UPLC Protein BEH C4 Column (300 Å, 2.1 × 50 mm, 1.7 μm). Mobile phase A was 0.1% formic acid in water and mobile phase B was acetonitrile with 0.1% formic acid. A constant flow

rate of 0.2 ml/min was used. Data was analyzed using Waters UNIFI software. Mass spectral deconvolution was performed using UNIFI software (version 1.9.4, Waters Corporation). The molecular weight of the protein was predicted using the ExPASy Compute pI/Mw tool, and chromophore maturation in GFP was also considered in the calculation.

**FACS analysis of amber suppression efficiency in mammalian cells.** For FACS analysis of live cells, HEK 293T cells were grown in 6-well plates (Corning) and co-transfected with pcDNA3.1 vector bearing aaRS/tRNA pair and pEGFP-mCherry-T2A-GFP-190TAG-His$_6$ in 1:1 ratio (μg: μg). Transfection was performed using lip2000 reagent (BioSharp) according to the manufacture's protocol with or without the corresponding UAA at the final concentration of 2 mM. After transfection (48 h), cells were trypsinized and taken up in full medium before centrifugation. Cells were centrifuged at 1400 g for 3 min, washed, and resuspended in PBS. FACS instrument (Beckman CytoFlex) was set up according to the manufacturer's instructions. HEK 293T cells were used to set appropriate forward scatter and side scatter gains. The fluorescent protein expressed cells were used to set FITC and PE gains and gate. At least 50,000 single cells were analyzed per condition. Lastly, GFP fluorescence was acquired at the FITC channel and mCherry fluorescence was acquired at the PE channel. FACS data were analyzed and processed with FlowJo (LLC).

**Engineering the chPheT in E. coli.** The tRNA library in pNEG vector carrying CAT-112TAG and GFP-190TAG dual reporter genes was transformed into DH10B competent cells harboring AzFRS-1. The transformed cells were recovered for 4 h at 37 °C then plated on LB agar containing 50 μg/ml kanamycin, 100 μg/ml ampicillin, 10 μg/ml chloramphenicol, and 0.2% L-arabinose in the presence of 2 mM AzF for 12 h at 37 °C and for additional 48 h at 30 °C. The survival clones with stronger fluorescence on each plate were picked. And full-length GFP fluorescence of these clones was further assayed in the presence or absence of AzF. The clones showing a significantly improved fluorescent signal over WT-chPheT were sequenced. These chPheT variants were further transformed into DH10B competent cells harboring the cognate chPheRS to verify the improvement of the amber suppression efficiency.

**Engineering the chPheRS in E. coli.** The C-terminal domain of AzFRS-1 was chosen as the target for gene diversification. The strategy for gene diversification was based on error-prone PCR[49]. Briefly, the chPheRS library from error-prone PCR was cloned into pBK vector, and electroporated into DH10B competent cells harboring positive selection plasmid pNEG carrying CAT-112TAG and GFP-190TAG dual reporter genes. The transformed cells were recovered for 4 h at 37 °C and then plated on LB agar containing 50 μg/ml kanamycin, 100 μg/ml ampicillin, 15 μg/ml chloramphenicol, and 0.2% L-arabinose in the presence of 2 mM AzF for 12 h at 37 °C and for additional 48 h at 30 °C. The clones with stronger fluorescence in the plate were picked, verified, and sequenced. The resultant chPheRS genes in each library were used as templates for the next round of error-prone PCR.

**Selection of the chPheRS variants for AzF in E. coli.** Positive selection was performed using the chPheRS library (F464NNK, T467NNK, and A507NNK), followed by negative selection, to identify the most active chPheRS variant for AzF incorporation. Briefly, the chPheRS library in pBK vector was firstly electroporated into DH10B competent cells harboring the positive selection plasmid pNEG-CAT-112TAG-chPheT-GFP-190TAG that contains CAT and GFP dual reporter genes with an amber codon (112TAG for CAT, 190TAG for GFP), respectively. The transformed cells were recovered for 4 h at 37 °C then plated on LB agar containing 50 μg/ml kanamycin, 100 μg/ml ampicillin, 10 μg/ml chloramphenicol, and 0.2% L-arabinose in the presence of 2 mM AzF for 12 h at 37 °C, followed by 48 h at 30 °C. After positive selection, the surviving pool was then electroporated into DH10B competent cells harboring the negative selection plasmid pNEG-Barnase-Q3TAG-D45TAG-chPheT in the absence of AzF. After two rounds of positive selections, the clones with fluorescence on the plate were picked and expressed in the presence or absence of 2 mM AzF. The fluorescence measurements were carried out as described above. Ten clones with the most prominent UAA-dependent GFP fluorescence were sent for sequencing.

**Construction of UAA-dependent synthetic auxotroph[50].** Using the dnaN gene as an example: The PKD4 plasmid was used as a PCR template to amplify kanamycin resistance cassette using the primers carrying 100-bp sequence homology for the chromosomal dnaN gene. The PCR production was purified by the DNA Clean & Concentrator™-5 (ZYMO Research). And then, ~1000 ng of purified PCR product was electroporated into the DH10B containing pTKRED and pChira-x*TAG-dnaN-AzFRS-2*chPheT. After electroporation, 1 ml 2xYT medium was added and cells were recovery in the presence of 1 mM AzF at 30 °C for 3 h. Cells were then centrifuged and plated on LB agar plates containing 30 μg/ml chloramphenicol, 50 μg/ml kanamycin, 100 μg/ml ampicillin, and 2 mM AzF at 30 °C for 48 h. The clones were picked up and grown in 200 μl LB medium containing required antibiotics and 2 mM AzF for 8 h. Clones were firstly verified by colony PCR across the desired insertion location, WT-DH10B was used as the negative control. The positive clones showed one band of ~1.6 kb that corresponds to the

ΔdnaN::KanR. These clones were washed three times in PBS and plated on LB agar plate in the presence or absence of 2 mM AzF at 37 °C for 2 days for verification.

**Assembly of orthogonal translation system integration cassette.** To generate an orthogonal 12D4-AzFRS-2 translation system integration cassette. The pChira-chPheRS-2*chPheT-site1 plasmid was used as a PCR template to amplify integration cassette using the primers carrying sequence homology. The PCR production was visualized 1% agarose gel stained with gel-red and correct size band was excised and purified using a gel extraction kit. And then, ~1000 ng of purified PCR product was electroporated into the DH10B strain containing pTKRED. Colonies were screened for correct integration by colony PCR and verified by Sanger sequencing. To generate a double AzF system integrated strain. The correct site1 integration strain was transformed pTKRED and prepared the competent cells. The integration cassette 2 was electroporated into the preceding strain and screen correct integration.

**Measurement of escape frequency.** The UAA-dependent synthetic auxotrophic cells were grown in the LB medium supplemented with 30 μg/ml chloramphenicol, 50 μg/ml kanamycin, 100 μg/ml ampicillin, and 2 mM AzF at 37 °C until OD$_{600}$ reaching to 0.6–0.8, then cells were washed three times in PBS and resuspended in 500 μl PBS. The washed cells were plated on LB agar medium containing 30 μg/ml chloramphenicol, 50 μg/ml kanamycin, 100 μg/ml ampicillin in the absence of AzF and grown at 37 °C for 14 days. Escape frequency was calculated as the escapes (c.f.u) per total cells plated.

**Growth curve analysis of bacterial strain.** Growth curve of DH10B strains bearing chPheRS, chPheT, or chPheRS/chPheT pair were determined by the plate reader (Bio Tek Synergy NEO2). Overnight grown cultures were diluted 1:100 with LB medium to a final volume of 1 ml. Bacteria were grown with orbital shaking (250 rpm) and OD$_{600}$ measurements were taken every 10 min in a 12-well plate. Growth was monitored for 650 min at 37 °C. Growth curve of UAA-dependent synthetic auxotrophs were determined with the use of the spectrophotometer DU730 (Beckman Coulter). Overnight grown cultures were diluted 1:100 with LB medium in the presence or absence of AzF to a final volume of 20 ml. Bacteria were grown with shaking (250 rpm) and OD$_{600}$ measurements were taken every 3 h. Growth was monitored for 39 h at 30 °C. The final OD$_{600}$ values were averaged for three biological replicates and were analyzed and processed by the Origin software.

**Time-course analysis of GFP expression by intact cell fluorescence measurements.** The DH10B strains harboring chromosomally integrated AzF translation system and GFP reporter plasmid were grown in a 24-well plate with LB medium supplemented with 30 μg/ml chloramphenicol, 50 μg/ml kanamycin, and 100 μg/ml ampicillin and allowed to grow at 37 °C to an OD$_{600}$ of 0.5–0.8. GFP expression was induced by the addition of 0.2% arabinose in the presence or absence of 1 mM AzF. The strains harboring AzF translation systems and GFP reporter plasmids were grown in 24-well plate with LB medium supplemented with 50 μg/ml kanamycin and 100 μg/ml ampicillin and allowed to grow at 37 °C to an OD$_{600}$ of 0.5–0.8. GFP expression was induced by the addition of 0.2% arabinose in the presence or absence of 1 mM AzF. GFP fluorescence was measured following expression with Bio Tek Synergy NEO2 every 80 s using excitation and emission wavelengths of 485 and 510 nm, respectively.

**Bioluminescence assay for UAA-dependent synthetic auxotroph.** To monitor the survival of UAA-dependent synthetic auxotroph in presence or absence of UAA, the plasmid pGEX-trp-fLuc was transformed into UAA-dependent synthetic auxotroph and plated onto LB agar containing 50 μg/ml kanamycin, 100 μg/ml ampicillin, 30 μg/ml chloramphenicol in the presence of 2 mM AzF. Next day, three clones were picked and grown in 2 ml LB medium containing required antibiotics and 2 mM AzF at 37 °C until OD$_{600}$ reaching ~0.8, then cell cultures were collected by centrifuging, and washed with 2 ml PBS four times. Subsequently, 20 μl of cells in PBS were inoculated in 2 ml LB medium with or without AzF, respectively. After 18 h, the cell cultures were collected by centrifuging and resuspended in 1 ml PBS. Each 100 μl suspension was transferred to a 96-well cell culture plate with addition of D-luciferin to the final concentration 200 μM. Bioluminescence signals were immediately captured with Bio Tek Synergy NEO2. DH10B cell transformed with plasmid pGEX-trp-fLuc was set as the positive control.

For imaging the bioluminescence of UAA-dependent synthetic auxotroph in living mice, Balb/c mice (male, 6–8 weeks) were purchased from Shanghai Model Organisms. The UAA-dependent synthetic auxotroph transformed with plasmid pGEX-trp-fLuc was treated as described above, and further resuspended in 0.2× LB medium to cultivate 4 h to deplete the remaining AzF. Cells (~1 × 10$^8$ cells/50 μL) were then subcutaneous injected into the mice (left legs, DH10B cell; right legs, UAA-dependent synthetic auxotroph with or without 5 mM AzF). AzF was subsequently injected at the final concentration of 10 mg/kg. Bioluminescence imaging in live mouse were performed at 24 h after injection. Luciferin (200 μL, 150 mg/kg) was administered into mice by intraperitoneal injection, and after another 10 min, the in vivo bioluminescence images of mice were acquired using IVIS Spectrum (FOV 23.4, f/1, Medium (8) bin, 60 s acquisition).

**Reporting summary**. Further information on research design is available in the Nature Research Reporting Summary linked to this article.

## Data availability
Any Supplementary Information (methods, figures, DNA sequences, and protein sequences) and chemical compound information are available in this paper. Source data are provided with this paper.

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

## Acknowledgements
We thank the National Key R&D Program of China (Grant 2019YFA09006600), the National Natural Science Foundation of China (Grant 91953113, 21877096), the Fundamental Research Funds for the Zhejiang Provincial Universities (Grant 2019XZZX003-19), and China Postdoctoral Science Foundation (2019M652072 for W.D.) for financial support. We are grateful to the core facility of Life Sciences Institute, Prof. Jie P. Li (Nanjing University) for supplying equipment for LC-MS analysis, Prof Yongzhen Xia (Shandong University) for sharing the pTKRed and PKD4 plasmids, and Dr. Vivian Y. Yu for helpful discussions.

## Author contributions
S.L. conceived the idea and supervised the study. H.Z. and W.D. conducted most experiments and analysed the data together. J.Z., L.H. and G.L. assisted with mice experiment. Y.Y. and Y.C. assisted with MS data analysis and interpretation. C.L. helped

with compound preparation. Y.F. helped with chromosome integration experiment. Y.Y. provided technical advice. S.L., H.Z. and W.D. wrote the manuscript. All authors commented on the final draft of the manuscript.

## Competing interests

The authors declare the following competing interests: S.L., H.Z., W.D., G.L. and Zhejiang University have filed a patent application (Chinese patent application number: 2021112939203) on the evolution and application of the chimeric phenylalanine system. All other authors declare no competing interests.
