## [Peer Review File · Nature Communications]

Reviewers' Comments:

Reviewer #1:

Remarks to the Author:

This manuscript reports an engineered chimeric aaRS system with an improved incorporation efficiency for unnatural amino acid. The utility of this system was demonstrated by constructing a synthetic auxotrophic bacterium. This work is solid and the chimeric aaRS system developed can potentially be useful. It can be published when the following comments are addressed.

1. For single-site incorporation studies, only one site (190 of GFP) was examined. It is necessary to study incorporation efficiency and fidelity at three or more different sites. In addition, it would be good to include another protein.
2. The authors should consider comparing their chimeric aaRS system to the previously reported AzFRS, including both efficiency and fidelity. The fidelity comparison is important since it is the key for the construction of synthetic auxotrophs.
3. The good fidelity was only observed when glns promoter was used. Why? A brief discussion should be included. It is not clear why high level of background was observed under other conditions.
4. If possible, it would be beneficial for others to use the system described in this work by providing plasmid maps and sequences in Supplementary Materials.

Reviewer #2:

Remarks to the Author:

The manuscript by Zhao et al. describes engineering of a chimeric phenylalanine system for efficient single- and multisite incorporation (amber codon-directed incorporation) of noncanonical amino acids into proteins with similar efficiencies as seen for wild-type protein expression, i.e. incorporation of natural, endogenous amino acids. This is shown for a series of Phe and Trp analogs. Furthermore, the authors use this engineered chimeric Phe system to create a series of customized E. coli strains where they introduced amber codon(s) in essential genes to construct synthetic auxotrophs that are dependent on the non-canonical amino acid.

The manuscript is a follow up from a nice paper the authors published in Nature Communications a year ago. The principle of the engineering and evolution of the different chimeric tRNA synthetase/tRNA systems combining synthetase parts of endogenous synthetases and pyrrolysyl tRNA synthetase (PylRS) together with PyltRNA was elaborated in this 2020 paper, including also a Phe-based chimeric system that showed 50% of wt-expression of GFP. In the present manuscript the authors evolved the chimeric Phe-tRNA to boost amino acid incorporation. Furthermore evolution of the N-terminal domain of the chimeric PheRS led to mutations outside of the amino acid binding pocket and to increased incorporation efficiencies of various Phe and Trp analogs into GFP. Also, multi-site incorporation of AzF into a GFP-fusion protein (ELP-GFP fusion) in an amber-KO strain was found to work with wt-efficiencies.

In the final part of the manuscript the authors use this engineered chimeric Phe system to construct synthetic E. coli auxotrophs whose growth tightly depends on addition of AzF by introducing amber codons into essential genes such as dnaN or adk.

Although the manuscript represents progress in developing more efficient systems for the site-specific incorporation of non-canonical amino acids and may therefore be useful to the community, I struggle to see the needed novelty in this manuscript to justify publication in Nat Commun. The chimeric Phe system was already developed in a previous publication from the authors. In this manuscript it was improved by (nowadays quite standard) directed evolution approaches of both the tRNA and the RS part. To really benchmark the newly evolved system and its efficiency it would have been important to benchmark it also against expression of proteins that are notoriously more difficult to amber-suppress than GFP. Is it possible to express other proteins containing non-canonical amino acids at wt-expression levels using this system? The construction

of synthetic auxotrophs using the evolved chimeric Phe systems seems a bit disconnected from the rest of the manuscript and a bit artificial and I struggled also here to see much novelty or foresee applications that cannot be done with other systems at hand.

We have taken into consideration of all the points brought up by the referees and addressed them with additional experiments and discussion. Our more detailed responses to each specific question raised by the referees are listed below. (For clarity, the original comments for each referee are shown in bold).

Reviewer #1:

This manuscript reports an engineered chimeric aaRS system with an improved incorporation efficiency for unnatural amino acid. The utility of this system was demonstrated by constructing a synthetic auxotrophic bacterium. This work is solid and the chimeric aaRS system developed can potentially be useful. It can be published when the following comments are addressed.

We really appreciate the referee for reviewing our paper and providing insightful comments.

1. For single-site incorporation studies, only one site (190 of GFP) was examined. It is necessary to study incorporation efficiency and fidelity at three or more different sites. In addition, it would be good to include another protein.

Thank you for pointing this out. In addition to site 190 on GFP, we have now randomly selected 7 other sites to study the incorporation efficiency and fidelity of our evolved system. Among them, 6 out of 7 sites showed wild-type like or surpassing efficiency in the GFP reporter assay (**Supplementary Figure 5**). And excellent incorporation fidelity (>99%) was detected by the LC-MS analysis of these GFP variants (**Supplementary Figure 5**). In addition, we have included eight more proteins that are more difficult to amber-suppress than GFP, including therapeutic protein, enzyme, structural protein, metalloprotein, and signaling protein, to demonstrate the broad utility of our system in various protein classes (**Figure 4C** and **Supplementary Figure 7**).

2. The authors should consider comparing their chimeric aaRS system to the previously reported AzFRS, including both efficiency and fidelity. The fidelity comparison is important since it is the key for the construction of synthetic auxotrophs.

Thank you for the suggestion. We have compared the efficiency and the background activity of our chimeric system with the previously reported AzFRS systems side-by-side (**Supplementary Figure 10**). Our system is the only one that exhibited wild-type like efficiency in the presence of UAA and extremely low background activity in the absence of UAA (**Supplementary Figure 10**). In addition, our evolved system can efficiently incorporate multi-site UAAs into ADK and DnaN proteins with undetectable background activity in DH10B cells, which was not possible using the previously reported AzFRS system (**Supplementary Figure 11**).

3. The good fidelity was only observed when glns promoter was used. Why? A brief discussion should be included. It is not clear why high level of background was observed under other conditions.

We apologize for the confusion. The fidelity of UAA incorporation by all the chimeric aaRS systems was verified by LC-MS analysis. For all the chimeric aaRS systems reported in our study, excellent fidelity (>99%) was observed (**Figure 3E and 4B**, and **Supplementary Figure 5, 7, and 8**). The high fidelity and efficiency of UAA incorporation by our evolved system is the key for the synthesis of UAA-containing protein for *in vitro* and *in vivo* application.

The referee is pointing out the background activity here. The background activity refers to the activity of aaRS/tRNA pair in the absence of UAA. Therefore, orthogonal translation system with low background activity and high efficiency is key for the construction of synthetic auxotrophs. Since we did not detect significant background activity in **Figure 2** and **Figure 5** using wild-type chimeric aaRS (**Figure 2D, E and 5A**), high background activity in Figure 3 may be due to the overexpression of highly active aaRS variants under the driven of strong promoter (**Figure 3C, D**). Similar results were observed using the previously reported *Mj*.AzFRS system as well (**Supplementary Figure 10** and **Nat Biotechnol 2015, 33(12): 1272-1279**). Therefore, reducing the expression level of highly active aaRS using a milder aaRS promoter (*glns*) resulted in much lower background activity (**Figure 3D**). We have now included this discussion in the revised main-text (**Page 8, text in red**).

4. If possible, it would be beneficial for others to use the system described in this work by providing plasmid maps and sequences in Supplementary Materials.

Thanks for pointing it out. We have now provided all plasmid maps and sequences in the supplementary materials.

Reviewer #2:

The manuscript by Zhao et al. describes engineering of a chimeric phenylalanine system for efficient single- and multisite incorporation (amber codon-directed incorporation) of noncanonical amino acids into proteins with similar efficiencies as seen for wild-type protein expression, i.e. incorporation of natural, endogenous amino acids. This is shown for a series of Phe and Trp analogs. Furthermore, the authors use this engineered chimeric Phe system to create a series of customized E. coli strains where they introduced amber codon(s) in essential genes to construct synthetic auxotrophs that are dependent on the non-canonical amino acid.

The manuscript is a follow up from a nice paper the authors published in Nature Communications a year ago. The principle of the engineering and evolution of the different chimeric tRNA synthetase/tRNA systems combining synthetase parts of endogenous synthetases and pyrrolysyl tRNA synthetase (PylRS) together with PyltRNA was elaborated in this 2020 paper, including also a Phe-based chimeric system that showed 50% of wt-expression of GFP. In the present manuscript the authors evolved the chimeric Phe-tRNA to boost amino

acid incorporation. Furthermore evolution of the N-terminal domain of the chimeric PheRS led to mutations outside of the amino acid binding pocket and to increased incorporation efficiencies of various Phe and Trp analogs into GFP. Also, multi-site incorporation of AzF into a GFP-fusion protein (ELP-GFP fusion) in an amber-KO strain was found to work with wt-efficiencies.

In the final part of the manuscript the authors use this engineered chimeric Phe system to construct synthetic *E. coli* auxotrophs whose growth tightly depends on addition of AzF by introducing amber codons into essential genes such as *dnaN* or *adk*.

We thank the reviewer for concisely summarizing our manuscript and commenting positively on our previous study.

1. Although the manuscript represents progress in developing more efficient systems for the site-specific incorporation of non-canonical amino acids and may therefore be useful to the community, I struggle to see the needed novelty in this manuscript to justify publication in Nat Commun. The chimeric Phe system was already developed in a previous publication from the authors. In this manuscript it was improved by (nowadays quite standard) directed evolution approaches of both the tRNA and the RS part. To really benchmark the newly evolved system and its efficiency it would have been important to benchmark it also against expression of proteins that are notoriously more difficult to amber-suppress than GFP. Is it possible to express other proteins containing non-canonical amino acids at wt-expression levels using this system?

We thank the referee for the helpful suggestion. In order to benchmark our newly evolved system and its efficiency, we have provided eight additional proteins that are more difficult to amber-suppress than GFP, including therapeutic protein, enzyme, structural protein, metalloprotein, and signaling protein (**Figure 4C and Page 8 and 9, text in red**). In the revised paper, purified protein yields of these UAA-containing proteins and wild-type proteins were obtained and listed for the side-by-side comparison (**Figure 4C**). The incorporation of UAA at these proteins was detected at a similar incorporation efficiency as natural amino acids, similar to the UAA incorporation efficiency on GFP-190TAG. And the extremely high incorporation fidelity of these UAA-containing proteins was verified by LC-MS (**Supplementary Figure 7**). In addition, we have randomly selected 7 sites other than site 190 on GFP to study the incorporation efficiency and fidelity of our evolved system in the suppression of amber codon with different location. Wild-type like or surpassing efficiency and excellent fidelity were observed, demonstrating the broad utility of our newly evolved system (**Supplementary Figure 5**).

2. The construction of synthetic auxotrophs using the evolved chimeric Phe systems seems a bit disconnected from the rest of the manuscript and a bit artificial and I struggled also here to see much novelty or foresee applications that cannot be done with other systems at hand.

First of all, having an orthogonal translation system that incorporating UAAs at the wild-type like efficiency and with low background activity in the absence of UAA is the crucial step for the construction of UAA-dependent synthetic auxotroph. In order to show the connection between the engineering part and the UAA-dependent synthetic auxotroph part of our study, we have now provided additional experiment to compare the efficiency and the background activity of our chimeric system with the previously reported *Mj.AzFRS* systems (**Supplementary Figure 10**). And our chimeric system shows 65.3-fold UAA-dependent protein expression and is the only one that showed wild-type like efficiency in the presence of UAA and extremely low background activity in the absence of UAA (**Figure 5A** and **Supplementary Figure 9 and 10**). We have shown in the revised paper that our evolved system can efficiently incorporate multi-site UAAs into essential protein such as ADK and DnaN proteins with undetectable background activity in DH10B cells (**Supplementary Figure 11**), which was not possible using the previously reported *Mj.AzFRS* system. Furthermore, we have re-written the manuscript to highlight the connections between these two parts (**Page 12, text in red**). Secondly, we have extended the application of UAA-dependent synthetic auxotroph in living mice (**Figure 6**), which has not been reported before according to our knowledge. Finally, because of its broad orthogonality in both prokaryotes and eukaryotes, the chimeric phenylalanine system can be used for the construction a series of UAA-dependent eukaryotic cells, which is the exciting frontier and is currently undergoing in our lab.

Reviewers' Comments:

Reviewer #1:

Remarks to the Author:

After reading their revised manuscript, this reviewer felt that the comments were addressed properly. The revised version is suitable for publication.

Reviewer #2:

Remarks to the Author:

The manuscript by Zhao et al. aims at improving amber suppression to be on par with wt-translation efficiencies. In a 2020 Nat Commun paper (doi: 10.1038/s41467-020-16898-y), the authors have shown that chimeric aaRS/tRNA systems, where both tRNA Synthetase (RS) and tRNA contain parts of endogenous aaRS/tRNA pairs and parts of the orthogonal pyrrolysine RS/tRNA pair are orthogonal to endogenous aaRS/tRNA pairs and can be used to efficiently incorporate unnatural amino acids (UAAs) into proteins. A couple of chimeric RS/tRNA systems that are orthogonal both in prokaryotic and eukaryotic cells has been evolved, among those a chimeric phenylalanine RS/tRNA pair (chPheRS/chPheT) that was able to incorporate phenylalanine in response to an amber codon in GFP with 50% efficiency compared to wt-GFP expression. Furthermore, two rationally introduced mutations in chPheRS (T467G and A507G = called AzFRS-1 in this manuscript) allowed to incorporate a variety of aromatic unnatural amino acids (AzF, NapA, 6MWR and 7MWR) with decent efficiencies.

In this manuscript, the authors aim to evolve this reported chimeric phenylalanine RS/tRNA pair (chPheRS/chPheT) for more efficient UAA incorporation. They start by evolving the tRNA through directed evolution approaches and find a tRNA variant (3C11-chPheT) that shows ca. 4-7-fold better incorporation yields for AzF and other UAAs than the progenitor chPheT. To further boost UAA incorporation efficiencies, the authors set out to improve the enzymatic activity of the chimeric AzFRS-1 through directed evolution by diversifying the C-terminal domain by error-prone PCR. They find a variant (12D4-AzFRS-1) that in combination with 3C11-chPheT shows 12-fold increased GFP fluorescence intensity upon expression of GFP-190TAG, equaling wt-GFP translation efficiencies. Unfortunately, also in the absence of any added UAA, equally strong fluorescence is observed, indicating that the evolved 12D4-AzFRS-1/3C11-chPheT pair shows a high degree of misincorporation of a natural amino acid and is highly unspecific. To decrease misincorporation, the authors create AzFRS-2 that contains one more mutation, but shows still very high degrees of protein expression in the absence of the UAA (ca 70-80%, Fig 3D)

The authors state that Trp is incorporated in the absence of any added UAA, but claim to show - by low-resolution full-length mass spec - that in the presence of AzF close to 100% incorporation of AzF is observed (they show such data for several TAG positions in GFP (Suppl. Fig 5) and for expression of different TAG-bearing proteins (Suppl. Fig 7)). As Trp and AzF have masses that only differ by 2 Daltons (Trp: 206 Da, AzF: 204 Da), low resolution ESI MS of full-length proteins is not suited to judge the fidelity of the evolved synthetase. To really show the fidelity of the evolved synthetase, the expressed proteins in absence and presence of UAA have to be expressed, and low-resolution full-length protein ESI-MS data have to be compared. Furthermore, tryptic digest and MS-MS data have to be shown. An alternative approach would be to label the AzF-containing proteins by Click-reaction with a (strained) alkyne and show quantitative shifts of the mass peaks or to show full-reduction of the azido group in AzF and thereby mass shift by 28 Da.

Given the fact that in the absence of UAA equal amounts of proteins are expressed, it would be highly surprising if the evolved synthetase was specific for UAA, when UAA is added. I assume that it is more likely that a mixture of proteins containing either AzF or a natural amino acid (Trp) is obtained and that the deconvolution program does not show two very close peaks but deconvolutes to one mass peak with an averaged mass. If the reported fidelities really hold true after showing the suggested experiments (i.e. full-length protein MS in presence and absence of UAA and MS-MS experiments and/or quantitative functionalization of AzF-bearing protein) this would be quite striking and there has to be an explanation as to why in absence and presence of UAA the same amount of protein is obtained but somehow in the presence of the UAA, the natural amino acid is not recognized anymore. The reason for this has to be shown, e.g. by in vitro amino acylation assays or similar.

In a second part of their manuscript the authors decide to use the evolved chimeric PheRS/tRNA

pair for creating auxotrophs. I suspect that due to the high unspecificity and misincorporation of the evolved 12D4 variant they have to go back to use AzFRS-2 (which is AzFRS-1 with one further mutation). They first compare incorporation efficiencies of this system with previously reported M.jannaschi AzFRS systems (Suppl Fig 10). Unfortunately, it is not clear what systems they use and this has to be better explained. I assume these are the originally published plasmid systems as mentioned in Suppl. Info references 1 and 2. If so, it has to be explained why Mj AzFRS-2 shows so much misincorporation in the absence of AzF. This has not been reported before. Using their newly evolved 3C11 tRNA and AzFRS-2 (which is in essence the same synthetase as used in the paper before containing one more mutation) the authors are able to create AzF-dependent bacterial auxotrophs by suppressing amber codons in essential genes such as dnaN and adk. This does not strike me as very novel application, as it has been reported multiple times before.

In light of the mentioned flaws and especially in lack of important experiments proving specificity and fidelity of the engineered synthetases I cannot recommend acceptance. The evolved 4D12 mutant seems to be highly unspecific and can therefore also not be used for the envisioned use of creating auxotrophs and the authors go back to use the previously published AzFRS-1 or AzFRS-2.

Since 4D12 is basically not usable because of its high misincorporation, an important experiment would be to show comparison of translation efficiencies of AzFRS-2 together with 3C11tRNA compared to the previous system i.e. AzFRS1 (only the two G mutations, but no F464I mutation) and the wt-chimeric Phe tRNA. This data is not included, but in my opinion presents the only novelty and progress compared to the previous manuscript.

Furthermore:

The manuscript is in parts not very well written and some paragraphs are difficult to understand. Furthermore, often information is missing. e.g in the first paragraph about evolution of chimeric tRNA it is not mentioned that this evolution is done in presence of AzF and with AzF-RS1 (which is the previously reported T467G/A507G chPheRS mutant). Equally, also for the evolution of chimeric aaRS it is not very clear what the starting synthetase is; I guess it must be AzF-RS1, the chimeric PheRS that contains the two G mutations mentioned above and incorporates AzF.

Also, for the directed evolution that leads to AzFRS-2 it is not clear if AzFRS-1 is the starting point or if this is done on the wt-chPheRS and if so, why?

Furthermore, it is often not immediately clear, in which experiments AzFRS-2 is used and where the 12D4 version of it is used.

Our more detailed responses to each specific question raised by the referees are listed below. (For clarity, the original comments for each referee are shown in bold).

Reviewer #1:

After reading their revised manuscript, this reviewer felt that the comments were addressed properly. The revised version is suitable for publication.

We really appreciate the valuable time the referee spent to evaluate our work and support our work for publication.

Reviewer #2:

1. The manuscript by Zhao et al. aims at improving amber suppression to be on par with wt-translation efficiencies. In a 2020 Nat Commun paper (doi: 10.1038/s41467-020-16898-y), the authors have shown that chimeric aaRS/tRNA systems, where both tRNA Synthetase (RS) and tRNA contain parts of endogenous aaRS/tRNA pairs and parts of the orthogonal pyrrolysine RS/tRNA pair are orthogonal to endogenous aaRS/tRNA pairs and can be used to efficiently incorporate unnatural amino acids (UAAs) into proteins. A couple of chimeric RS/tRNA systems that are orthogonal both in prokaryotic and eukaryotic cells has been evolved, among those a chimeric phenylalanine RS/tRNA pair (chPheRS/chPheT) that was able to incorporate phenylalanine in response to an amber codon in GFP with 50% efficiency compared to wt-GFP expression. Furthermore, two rationally introduced mutations in chPheRS (T467G and A507G = called AzFRS-1 in this manuscript) allowed to incorporate a variety of aromatic unnatural amino acids (AzF, NapA, 6MWR and 7MWR) with decent efficiencies.

We are very grateful to the referee for the valuable time evaluating our work again. In our previous manuscript, the progenitor chPheRS/tRNA pair incorporated most UAAs with approximately 3-20% efficiency compared to the wild-type protein. In this manuscript, we observed that the engineered chPheRS/tRNA pair incorporated these UAAs with efficiency similar to that of wild-type.

2. In this manuscript, the authors aim to evolve this reported chimeric phenylalanine RS/tRNA pair (chPheRS/chPheT) for more efficient UAA incorporation. They start by evolving the tRNA through directed evolution approaches and find a tRNA variant (3C11-chPheT) that shows ca. 4-7-fold better incorporation yields for AzF and other UAAs than the progenitor chPheT. To further boost UAA incorporation efficiencies, the authors set out to improve the enzymatic activity of the chimeric AzFRS-1 through directed evolution by diversifying the C-terminal domain by error-prone PCR. They find a variant (12D4-AzFRS-1) that in combination with 3C11-chPheT shows 12-fold increased GFP fluorescence intensity upon expression of GFP-190TAG, equaling wt-GFP translation efficiencies. Unfortunately, also in the absence of any added UAA, equally strong fluorescence is observed, indicating that the evolved 12D4-AzFRS-

1/3C11-chPheT pair shows a high degree of misincorporation of a natural amino acid and is highly unspecific. To decrease misincorporation, the authors create AzFRS-2 that contains one more mutation, but shows still very high degrees of protein expression in the absence of the UAA (ca 70-80%, Fig 3D). The authors state that Trp is incorporated in the absence of any added UAA, but claim to show - by low-resolution full-length mass spec - that in the presence of AzF close to 100% incorporation of AzF is observed (they show such data for several TAG positions in GFP (Suppl. Fig 5) and for expression of different TAG-bearing proteins (Suppl. Fig 7)). As Trp and AzF have masses that only differ by 2 Daltons (Trp: 206 Da, AzF: 204 Da), low resolution ESI MS of full-length proteins is not suited to judge the fidelity of the evolved synthetase. To really show the fidelity of the evolved synthetase, the expressed proteins in absence and presence of UAA have to be expressed, and low-resolution full-length protein ESI-MS data have to be compared. Furthermore, tryptic digest and MS-MS data have to be shown. An alternative approach would be to label the AzF-containing proteins by Click-reaction with a (strained) alkyne and show quantitative shifts of the mass peaks or to show full-reduction of the azido group in AzF and thereby mass shift by 28 Da.

Thank you for pointing this out. As suggested, we have performed the reduction reaction on the azido group on all GFP variants. The GFP variants carrying AzF were expressed and purified using the 12D4-AzFRS-2/3C11-chPheT pair. The molecular weight (MW) of these GFP variants before and after tris(2-carboxyethyl)phosphine (TCEP) treatment were examined by LC-MS analysis. The results showed that the azido group on all GFP variants were completely reduced by TCEP, demonstrating the high incorporation fidelity of AzF by the 12D4-AzFRS-2/3C11-chPheT pair in the presence of AzF (**Supplementary Figure 5**). In addition, our ESI-TOF mass spectrometry can distinguish to some extent between GFP variants with only 2 Daltons differences and therefore between Trp and AzF on the protein (Trp: 204Da, AzF: 206Da). The observed and expected MW of GFP-190AzF were 27797 Da and 27796 Da (**Figure 3E**), while the observed and expected MW of GFP-190Trp were 27795 Da and 27794 Da, respectively (**Supplementary Figure 5B**). For AzF-incorporated proteins, we typically detected the same MW or +1 Da MW as expected (**Supplementary Figures 5**), also suggesting the high incorporation fidelity of AzF by the 12D4-AzFRS-2/3C11-chPheT pair. Furthermore, we have demonstrated that 12D4-chPheRSs have exhibited extremely high fidelity to incorporate a variety of UAAs, including AcF, NapA, 6MW, 7MW, and BTA (**Figure 4A-B**). And no misincorporation of Trp was detected in these protein samples (**Figure 4A-B**). Because the beneficial mutations of 12D4-chPheRS do not present in the amino acid binding pocket, the incorporation fidelity should be translated to all UAAs, including AzF. Together, these results demonstrated that 12D4-AzFRS-2 incorporated AzF with excellent fidelity.

3. Given the fact that in the absence of UAA equal amounts of proteins are expressed, it would be highly surprising if the evolved synthetase was specific for UAA, when UAA is added. I assume that it is more likely that a mixture of proteins

containing either AzF or a natural amino acid (Trp) is obtained and that the deconvolution program does not show two very close peaks but deconvolutes to one mass peak with an averaged mass. If the reported fidelities really hold true after showing the suggested experiments (i.e. full-length protein MS in presence and absence of UAA and MS-MS experiments and/or quantitative functionalization of AzF-bearing protein) this would be quite striking and there has to be an explanation as to why in absence and presence of UAA the same amount of protein is obtained but somehow in the presence of the UAA, the natural amino acid is not recognized anymore. The reason for this has to be shown, e.g. by *in vitro* amino acylation assays or similar.

Thank you for your suggestion. We used the *in vivo* kinetic analysis assay to explain its high incorporation fidelity in the presence of UAA. The production of GFP variants in the presence or absence of UAA was monitored in real time in intact cells by the *in vivo* kinetic analysis assay, which was previously used to investigate the selectivity mechanism *in vivo* (*Nat. Biotechnol.*, **2015**, 33, 1272-1279, Figure S10-11). Time-course analysis of the expression of GFP-190TAG and GFP-2*TAGs using 12D4-AzFRS-2 revealed a significantly reduced rate of protein production in the absence of AzF in living cells (**Supplementary Figure 8**), suggesting that 12D4-AzFRS-2 kinetically favored AzF over Trp. Therefore, we hypothesized that the high background activity was likely generated by overexpression of the highly active plasmid-based 12D4-AzFRS variants, and a series of experiments were designed to verify this. Indeed, when 12D4-AzFRS-2 was integrated into the chromosome, background activity was significantly reduced and excellent amber suppression efficiency was still observed (**Supplementary Figure 7-8**). Similarly, when the expression of 12D4-AzFRS-2 was driven by the mild glutaminyl-tRNA synthetase (glns) promoter rather than the strong promoter *oxb20* in the plasmid, the background activity of 12D4-AzFRS-2 was significantly reduced and high amber suppression efficiency was still observed (**Figure 3D**). Together, these results demonstrated that 12D4-chPheRS kinetically favored UAAs over NAAs and the high background activity was caused by plasmid-based overexpression of 12D4-chPheRS. We have now included this discussion in the revised main-text.

Furthermore, similar results were observed in the previous publication (*Nat. Biotechnol.*, **2015**, 33, 1272-1279, Figure 3 and Figure S10). The author also detected high background activity using the highly active AzFRS.2.t1 (Mj.AzFRS-2 in our study). However, the *in vivo* kinetic analysis assay showed that AzFRS.2.t1 kinetically favored AzF over NAAs (*Nat. Biotechnol.*, **2015**, 33, 1272-1279, Figure 3 and Figure S10).

Nat. Biotechnol., 2015, 33, 1272-1279. **Figure 3**

Figure 10. Kinetic analysis of GFP(3UAG) production by AARS variants expressed on plasmids

Nat. Biotechnol., 2015, 33, 1272-1279. **Figure S10**

4. In a second part of their manuscript the authors decide to use the evolved chimeric PheRS/tRNA pair for creating auxotrophs. I suspect that due the high unspecificity and misincorporation of the evolved 12D4 variant they have to go back to use AzFRS-2 (which is AzFRS-1 with one further mutation). They first compare incorporation efficiencies of this system with previously reported M.jannaschi AzFRS systems (Suppl Fig 10). Unfortunately, it is not clear what

systems they use and this has to be better explained. I assume these are the originally published plasmid systems as mentioned in Suppl. Info references 1 and 2. If so, it has to be explained why Mj AzFRS-2 shows so much misincorporation in the absence of AzF. This has not been reported before. Using their newly evolved 3C11 tRNA and AzFRS-2 (which is in essence the same synthetase as used in the paper before containing one more mutation) the authors are able to create AzF-dependent bacterial auxotrophs by suppressing amber codons in essential genes such as *dnaN* and *adk*. This does not strike me as very novel application, as it has been reported multiple times before.

Thank you for the suggestion. We have revised the text to clarify that Mj.AzFRS-2 is the previously published AzFRS.2.t1 (*Nat. Biotechnol.*, **2015**, *33*, 1272-1279.) (Supplementary Figure 13A). In contrast, AzFRS.2.t1 was reported to exhibit very high background activity when overexpressed in the plasmid (*Nat. Biotechnol.*, 2015, *33*, 1272-1279. Figure 3 and Figure S10). Please refer to the attached figures under Question #3. The author detected high background activity of GFP-3*TAGs in the absence of UAA when the highly active AzFRS.2.t1 was overexpressed in the plasmid. Low background activity was detected when Mj.AzFRS-2 was integrated into the chromosome (*Nat. Biotechnol.*, **2015**, *33*, 1272-1279.). Similar results were observed in our study when using chromosomally integrated 12D4-AzFRS-2 and/or a weaker aaRS promoter (Supplementary Figure 7-8 and Figure 3D).

Although essential genes such as *dnaN* and *adk* have been used to generate synthetic auxotroph, multiple amber codons must be inserted into multiple essential genes simultaneously to achieve low escape frequencies in the genome recorded *E. coli* (*Nature*, **2015**, *518*, 55.). Our study provided a generalizable approach to engineer synthetic auxotroph from a common *E. coli* strain with much lower escape frequency by inserting amber codons into one essential gene. And we have extended the application of UAA-dependent synthetic auxotroph in living mice.

5. In light of the mentioned flaws and especially in lack of important experiments proving specificity and fidelity of the engineered synthetases I cannot recommend acceptance.

As suggested, we have performed additional experiments to demonstrate the high fidelity of AzF incorporation. And we have used the *in vivo* kinetic analysis assay to explain its high incorporation fidelity.

6. The evolved 4D12 mutant seems to be highly unspecific and can therefore also not be used for the envisioned use of creating auxotrophs and the authors go back to use the previously published AzFRS-1 or AzFRS-2. Since 4D12 is basically not usable because of its high misincorporation, an important experiment would be to show comparison of translation efficiencies of AzFRS-2 together with 3C11tRNA compared to the previous system i.e. AzFRS1 (only the two G mutations, but no F464I mutation) and the wt-chimeric Phe tRNA. This data is not included, but in my opinion presents the only novelty and progress compared to the previous manuscript.

As suggested, we have compared the efficiency and background activity of these mentioned AzFRS systems, including AzFRS-1, AzFRS-2, and 12D4-AzFRS-2, with the progenitor pair (**Supplementary Figure 13B**). The engineered AzFRS-2/3C11-chPheT pair showed greatly improved activity (~10-fold) compared to the progenitor pair. Besides, AzFRS-2/3C11-tRNA pair can efficiently incorporate AzF to various sites of GFP with 60-95% efficiency compared to WT-GFP, with extremely low background activity (**Supplementary Figure 6**). Therefore, 12D4-AzFRS-2/3C11-tRNA pair can be used for efficient production of protein material and AzFRS-2/3C11-tRNA pair can be used for generating UAA-dependent synthetic auxotrophs.

7. Furthermore: The manuscript is in parts not very well written and some paragraphs are difficult to understand. Furthermore, often information is missing. e.g in the first paragraph about evolution of chimeric tRNA it is not mentioned that this evolution is done in presence of AzF and with AzF-RS1 (which is the previously reported T467G/A507G chPheRS mutant). Equally, also for the evolution of chimeric aaRS it is not very clear what the starting synthetase is; I guess it must be AzF-RS1, the chimeric PheRS that contains the two G mutations mentioned above and incorporates AzF.

With the help of a native speaker, we proofread the grammar of our manuscript and highlighted the major changes in red. And we also edited the text and method section to clarify that we used AzFRS-1 for the directed evolution of chimeric tRNA and chimeric PheRS.

8. Also, for the directed evolution that leads to AzFRS-2 it is not clear if AzFRS-1 is the starting point or if this is done on the the wt-chPheRS and if so, why?

We have made it clear in the text and methods that WT-chPheRS was used as the starting point to generate AzFRS-2, in order to generate a more diverse library from which to find the most active chPheRS variant for AzF incorporation.

9. Furthermore, it is often not immediately clear, in which experiments AzFRS-2 is used and where the 12D4 version of it is used.

In the revised paper, when 12D4 version was used, we have added the prefix, 12D4-, before chPheRS to make it clearer. Thank you for pointing it out.

Reviewers' Comments:

Reviewer #2:

Remarks to the Author:

This is already the third version of this manuscript and I still struggle to see the required conceptual novelty that would justify publication in Nat Commun.

I really liked the 2020 Nat Commun paper by the same authors that describes the development of hybrid aaRS/tRNA pairs for efficient incorporation of aromatic non-canonical amino acids. This is a conceptually very interesting and innovative work!

The manuscript under question represents however only an incremental progress over this 2020 Nat Commun paper. The authors first improve the tRNA component (3C11) of the hybrid aaRS/tRNA pair and then do evolution on the C-terminal domain, which leads to the 12D4-RS/3C11 aaRS/tRNA version that shows efficient incorporation of AzF, but produces the same amount of protein in the absence of the UAA. The authors show mass-spec data for purified proteins expressed with the 12D4-RS/3C11 pair that match the calculated masses and they argue that the faster kinetics of the evolved pair towards UAAs compared to natural AAs accounts for the specificity, but the provided in vivo kinetics data (Suppl. Fig 8A: 12D4-AzFRS2/3C11 in plasmid with GFP-190TAG) do not really do this justice as they show at any time point at least 50% of GFP-fluorescence arising from mis-incorporation of natural amino acids.

Integrating the 12D4-RS into the genome or using weaker promoters helps in reducing misincorporation but this comes with somewhat reduced expression yields, similar to those ones achieved by using just the AzFRS-2 synthetase together with the evolved 3C11 tRNA.

Therefore it is not clear to me what the advantage of the 12D4-variant should be. This is also shown by the fact that for the envisioned use – creating of auxotrophs – the authors have to use the AzFRS2/3C11 synthetase and the 12D4 mutations outside of the amino acid binding pocket – that would make the approach more universal, because it could be applied to many different synthetases for various UAAs – are not really useful.

In light of this - in my personal opinion - the conceptual novelty and general impact for the genetic code expansion field is a bit vague and does not justify publication in Nat. Communications.

Our more detailed responses to each specific question raised by the referees are listed below. (For clarity, the original comments for each referee are shown in bold).

Reviewer #2 (Remarks to the Author):

This is already the third version of this manuscript and I still struggle to see the required conceptual novelty that would justify publication in Nat Commun.

I really liked the 2020 Nat Commun paper by the same authors that describes the development of hybrid aaRS/tRNA pairs for efficient incorporation of aromatic non-canonical amino acids. This is a conceptually very interesting and innovative work!

In light of this - in my personal opinion - the conceptual novelty and general impact for the genetic code expansion field is a bit vague and does not justify publication in Nat. Communications.

First of all, we would like to thank you for your helpful comments that have greatly improved our manuscript. Our work presents broadly orthogonal translation systems that allow single-site and multi-site incorporation of various UAAs with efficiencies similar to that of NAAs into model proteins and several functional proteins. The incorporation of UAAs with wild-type like efficiency is extremely challenging and has rarely been reported. Furthermore, we report a generalizable way to construct synthetic auxotroph and apply it to live animals, which in our opinion is also very concise and novel application.

The manuscript under question represents however only an incremental progress over this 2020 Nat Commun paper. The authors first improve the tRNA component (3C11) of the hybrid aaRS/tRNA pair and then do evolution on the C-terminal domain, which leads to the 12D4-RS/3C11 aaRS/tRNA version that shows efficient incorporation of AzF, but produces the same amount of protein in the absence of the UAA.

In our previous manuscript, the progenitor chPheRS/tRNA pair incorporated most UAAs with approximately 3-20% efficiency compared to the wild-type protein. In this manuscript, we observed that the engineered chPheRS/tRNA pair incorporated these UAAs with efficiency similar to that of wild-type. The incorporation of UAAs with wild-type like efficiency is extremely challenging and has rarely been reported.

When using 12D4-AzFRS-2/3C11 system, excellent AzF incorporation fidelity (>99%) at several sites of GFP before and after TCEP treatment was detected by LC-MS analysis (Figure S5). These data demonstrated that 12D4-AzFRS-2/3C11 system can efficiently incorporate AzF with an activity on par with that of natural amino acids and excellent fidelity (Figure 3d, 4a, and S5). Although the background activity of 12D4-AzFRS-2 was relatively high in the absence of AzF (Figure 3d), the background activity detected in the absence of AzF was about 60-70% of that of the experimental group in the presence of AzF (Figure 3d and S5). This (60-70% of background activity) is very different from the comment that “**12D4-RS/3C11 aaRS/tRNA produces the same amount of protein in the absence of UAA**”. In fact, although the activity of 12D4-chPheRS was about 6-fold higher than that of

chPheRS (Figure 3a), we could only increase the AzF incorporation efficiency from 60-80% (Figure S6) to 90-110% (Figure S5) of WT GFP when using 12D4-AzFRS-2, which was 6-fold more active than AzFRS-2 (Figure 3a and S5-6). Therefore, in the absence of AzF, the background activity (60-70%) is significantly lower than that in the presence of AzF. Most importantly, we can incorporate several UAAs with wild-type like efficiency and excellent fidelity into several functional proteins (Figure 3d-e, 4a-c, S4-5, and S13). Moreover, we have reported several orthogonal translation systems with wild-type like efficiency and extremely low background activity as well (Supplementary Figure 6-7).

The authors show mass-spec data for purified proteins expressed with the 12D4-RS/3C11 pair that match the calculated masses and they argue that the faster kinetics of the evolved pair towards UAAs compared to natural AAs accounts for the specificity, but the provided in vivo kinetics data (Suppl. Fig 8A: 12D4-AzFRS2/3C11 in plasmid with GFP-190TAG) do not really do this justice as they show at any time point at least 50% of GFP-fluorescence arising from mis-incorporation of natural amino acids.

Integrating the 12D4-RS into the genome or using weaker promoters helps in reducing misincorporation but this comes with somewhat reduced expression yields, similar to those ones achieved by using just the AzFRS-2 synthetase together with the evolved 3C11 tRNA.

Therefore it is not clear to me what the advantage of the 12D4-variant should be. This is also shown by the fact that for the envisioned use – creating of auxotrophs – the authors have to use the AzFRS2/3C11 synthetase and the 12D4 mutations outside of the amino acid binding pocket – that would make the approach more universal, because it could be applied to many different synthetases for various UAAs – are not really useful.

The kinetic activity of the 12D4-AzFRS-2/3C11 system in the presence and absence of AzF cannot be compared under enzyme saturation conditions when aaRS was strongly overexpressed. Therefore, genomic integration and the use of weaker promoter were intended to create a condition for kinetic activity comparisons. There is a misunderstanding of the rationale for the experiment.

In order to investigate the excellent fidelity of 12D4-AzFRS-2/3C11 system, we performed MS experiments (Figure S5), side-by-side comparison experiments (Figure S6-7 and S13) and in vivo kinetics analysis assays (Figure 7-8) in the presence and absence of AzF, as suggested. For the in vivo kinetic analysis experiments, the activity of 12D4-AzFRS-2 was easily saturated in the presence of AzF when aaRS was strongly overexpressed. For example, when integrated into genome, the efficiency was 90% in the presence of AzF (Figure S7), whereas when plasmid-based overexpression was used, the activity was 110% (Figure 3D). In contrast, the activity of 12D4-AzFRS-2 did not saturate in the absence of AzF, whereas it could be significantly increased when aaRS was strongly overexpressed on plasmid. For example, when integrated into genome, the activity was 15% in the absence of AzF (Figure S7), whereas when plasmid-based overexpression was used, the activity was

70% (Figure 3D), which was more than 4-fold higher. Therefore, plasmid-based overexpression contributed to the relatively high background activity. And kinetic activity of the 12D4-AzFRS-2/3C11 system in the presence or absence of AzF cannot be compared under enzyme saturation conditions with strong overexpression of aaRS. To compare the kinetic activity of the 12D4-AzFRS-2/3C11 system in the presence and absence of AzF, we integrated 12D4-AzFRS-2/3C11 pair into genome (Figure S7 and S8c) or used weaker promoters (Figure 3d and S8a-b) in order to reduce the expression level of aaRS, when the activity of 12D4-AzFRS-2 didn't reach saturation, allowing for parallel kinetic comparisons. We found that in kinetic activity of 12D4-AzFRS-2/3C11 pair was near 10-fold higher in the presence of AzF, as determined by measuring the slope of GFP production curve at different time points (Figure S7 and S8c). And the kinetic properties of 12D4-AzFRS-2/3C11 pair should not altered when 12D4-AzFRS-2 was overexpressed. Therefore, 12D4-AzFRS-2 kinetically favored UAAs over NAAs. Even when we used plasmid-based overexpression of 12D4-AzFRS-2, the kinetic activity in the presence of AzF was still 3-5 fold higher than in the absence of AzF by measuring the slope of GFP production curve (Figure S8a).

Together, these data strongly demonstrated that 12D4-AzFRS-2 kinetically favored UAAs over NAAs and the high background activity was caused by plasmid-based overexpression of 12D4-chPheRS. Furthermore, our results were consistent with the previous report (Nat Biotechnol 2015, 33(12): 1272-1279. Figure S10E-F). The previous report used GFP-3*TAG production to investigate the kinetic effects. As the number of TAG codons in the GFP reporter increased, larger differences were detected in the slope of GFP production curve (S8a-b). Impressively, our 12D4-AzFRS-2/3C11 system kinetically favored UAAs over NAAs, even using the GFP reporter with one TAG codon (Figure S8a, c-d) or two TAG codons (Figure S8b).

Taken together, 12D4-chPheRSs are among the most efficiency system for UAAs incorporation with high fidelity of incorporation.

Figure 10. Kinetic analysis of GFP(3UAG) production by AARS variants expressed on plasmids

Nat. Biotechnol., 2015, 33, 1272-1279. **Figure S10**